# Intense and localized export of selected marine snow types at eddy edges in the South Atlantic Ocean

Alexandre Accardo[1], Rémi Laxenaire[4,5,6], Alberto Baudena[1], Sabrina Speich[3,4], Rainer Kiko[2], Lars Stemmann[1,7]

[1]Sorbonne Université, CNRS, Laboratoire d'Océanographie de Villefranche, UMR 7093 LOV, Villefranche-sur-Mer, France
[2]GEOMAR Helmholtz Center for Ocean Research Kiel, Germany
[3]Ecole Normale Supérieure, PSL Research University, France
[4]UMR 8539 Laboratoire de météorologie dynamique (LMD), France
[5]UMR 8105 Laboratoire de l'Atmosphère et des Cyclones, Saint-Denis de La Réunion, France
[6]Center for Ocean-Atmospheric Prediction Studies, Florida State University, Tallahassee, FL, United States
[7]Institut Universitaire de France (IUF), France

*Correspondence to*: Alexandre Accardo (alexandre.accardo@imev-mer.fr)

## Abstract

The Biological Carbon Pump (BCP) comprises a wide variety of processes involved in transferring organic carbon from the surface to the deep ocean. This results in long-term carbon sequestration. Without BCP, atmospheric $CO_2$ concentrations would be around 200 ppm higher. This study reveals that ocean dynamics at mesoscale and submesoscale could have a major impact on particulate organic matter (POM) vertical distribution. Our results suggest that intense submesoscale frontal regions, such as those between mesoscale eddies, could lead to an important accumulation and transport of particulate organic matter (POM) from the mixed layer depth (MLD) down to the mesopelagic zone. To reach these conclusions, a multifaceted approach was applied. It included in-situ measurements and marine snow images from a BGC Argo float equipped with an Underwater Vision Profiler (UVP6), satellite altimetry data, and Lagrangian physics diagnostics. We focused our study on three intense features in marine snow distribution observed during the 17-month long float mission in the Cape Basin, southwest of Africa. These features were located in the frontal region between mesoscale eddies. Our study suggests that a particle injection pump induced by a frontogenesis-driven mechanism has the potential to enhance the effectiveness of the biological pump by increasing the depth at which carbon is injected into the water column. This work also emphasizes the importance of establishing repeated sampling campaigns targeting the interface zones between eddies. This could improve our understanding of the mechanisms involved in the deep accumulation of marine snow observed at eddy interfaces.

## 1 Introduction

The open ocean plays a critical role in mitigating climate change by storing carbon dioxide ($CO_2$) away from the atmosphere for long periods of time (Boyd et al., 2019). This carbon storage occurs over decades to thousands of years and relies on two well-established mechanisms that create a gradient of dissolved inorganic carbon (DIC) between the surface and deep regions of the ocean and enhances DIC concentrations at depth. These mechanisms are known as the biological pump and the solubility pump (or physical pump). The solubility pump transports cold, dense, and DIC-rich waters to the deep ocean, as part of the ocean's overturning circulation, primarily in high-latitude regions (Sarmiento and Gruber, 2006). The biological carbon pump (BCP), on the other hand, involves the export of particulate organic carbon (POC) from surface waters below the euphotic depth, and operates worldwide. The BCP encompasses a wide range of processes, from the fixation of atmospheric $CO_2$ by phytoplankton activity, to carbon sequestration into the deep ocean (Le Moigne, 2019). The BCP contributes also to approximately 90% of the dissolved inorganic carbon (DIC) gradient between surface and deep ocean (Boyd et al., 2019). Without the BCP atmospheric concentrations of the greenhouse gas $CO_2$ would be approximately 200 ppm higher than in a world without biology (Maier-Reimer et al. 1996) and the global climate would be much warmer by default.

A key component of the BCP is the biological gravitational pump (BGP). The BGP is the process by which large and/or dense aggregates of POC sink due to gravity. Large particles of POC (> 500 µm), also called marine snow, can consist of aggregated phytoplankton cells such as diatoms, different types of aggregates resulting from coagulation processes (Le Moigne, 2019), and zooplankton activity, such as phytoplankton grazing and faecal pellet egestion (Turner, 2015). This mechanism is estimated, through empirical and food web models, to export about 3 to 10 Pg of carbon per year below the euphotic zone or 100 meters depth depending on the study (Dunne et al., 2005; Henson et al., 2012; Siegel et al., 2014; Bisson et al., 2020). The speed at which surface aggregates sink to the deep ocean depends on their individual sinking rates, which are influenced by their combined morphological (size, porosity, etc.) and chemical properties (ratio of organic versus mineral, mineral type, etc.) (Stemmann et al., 2004 ; Cael et al., 2021 ; Trudnowska et al., 2021 ; Soviadan et al., submitted). Laboratory experiments have estimated that the sinking velocity of marine snow range from a few meters to several hundred meters per day (Shanks and Trent, 1980; Azetsu-Scott and Johnson, 1992; Ploug et al., 2008; Laurenceau-Cornec et al., 2015; Cael et al., 2021) and, in a few cases, from in situ imaging time series (Alldredge and Gotschalk, 1988; Stemmann et al., 2002; Trudnowska et al., 2021; Soviadan et al., 2024). In situ imagery has proven to be a valuable tool for estimating sinking speeds. This can improve our understanding of particle dynamics, which affect the BGP.

Another important component of the BCP that can occur concurrently with the BGP, are the particle injection pumps (PIPs). These consist of the export of carbon mediated by plankton migrations or physical vertical movement and can also play a significant role in the ocean's capacity to store carbon. PIPs encompass a variety of mechanisms, temporal-spatial scales, and geographical extent, and can affect all types of particles in the water column. They typically transport particles below the euphotic zone. Depending on the specific injection mechanism, PIPs can reach depths greater than 1000 m (Boyd et al. 2019).

The physical processes that make up the PIPs include subduction resulting from the mixed-layer shallowing (referred to as the
mixed-layer pump), subduction caused by the large-scale ocean circulation over distances of 100-1000 km (referred to as the
large-scale subduction pump), and subduction mediated by mesoscale (10-100 km, few weeks to several months) to
submesoscale (100 m to 10 km, hours to days) ocean dynamics (Thomas et al., 2008; McWilliams, 2016).
In particular, the influence of mesoscale eddies on particle production, aggregation and export (Stemmann et al., 2008 ; Guidi
et al., 2012 ; McGillicuddy, 2016 ; Stuckel et al., 2017) is a very active area of research because these structures are ubiquitous
in the global ocean and are the largest source of ocean variability (Stammer, 1997; Wunsch, 1999). For example, spatial
patterns of particles, differentiated by size, have been shown to be associated with anticyclonic circulation (Gorsky et al., 2002;
Waite et al., 2016). Upwelling at eddy's interfaces can enhance phytoplankton productivity and particle production (Shih et
al., 2020) while downwelling, contributes to oblique transport of dissolved, and particulate (both sinking and suspended)
carbon to depth (Stemmann et al., 2008; Guidi et al., 2012) in a process called the Eddy Subduction Pump (ESP; Omand et al.
2015). Vertical transport has been suggested to be increased by sub-mesoscale vertical velocities associated with fronts (Guidi
et al., 2007). Understanding the mechanisms associated with physical-biological biogeochemical interaction is challenging due
to the transient nature of the underlying processes. Therefore, it is essential to employ multidisciplinary approaches, including
in situ observations, to elucidate these intricate phenomena (McGillicuddy, 2016).
This work is based on the deployment of a new prototype of BGC Argo float (part of the Argo international program:
https://argo.ucsd.edu) equipped with an Underwater Vision Profiler 6 (UVP6; Picheral et al., 2022) and several physical-
biogeochemical sensors in the Southeast Atlantic Ocean. The float was deployed in the Cape Basin, a region southwest of
Africa, known for its intense eddy activity, which is mainly controlled by a complex interaction between the Benguela
Upwelling, the Agulhas Current System and the South Atlantic Current (Richardson et al., 2003; Boebel et al., 2003). The
float was recovered after one year of operation and the collected images were analyzed and classified into plankton categories
and morphological types of marine snow (Trudnowska et al., 2022). Hydrological and particle data collected by the float were
then combined with satellite altimetry to identify frontal structures and mesoscale eddies (Chaigneau et al., 2009; Laxenaire et
al., 2018; 2019; 2020; 2024). Based on recurrent observations of marine snow concentration "hot spots" at mesopelagic depth
during the productive season, we discuss the role of horizontal and vertical circulation at fronts on marine snow production
and export to the deep sea.
**2 Material and Methods**
**1.1 Observing strategy**
During the SO283 cruise, a BGC Argo float (WMO: 6903095) was deployed, from the RV Sonne, on April 14, 2021, in a
cyclonic eddy in the southeastern region of the Atlantic Ocean, near the Benguela upwelling system at 33.0978°S and

13.8673°E (Fig. 1.A). The float remained within the eddy for about 5 months. During this period, the eddy merged with another cyclone (Baudena et al., 2023), until it disappeared from satellite altimetry maps probably due to subduction (Ioannou et al., 2022). A detailed analysis of this timeframe of the dataset is provided in Baudena et al., (2023). It then sampled different mesoscale features within the region, and after a sampling period of about 1.5 years, it was retrieved by the S.A. Agulhas II on September 19, 2022. During its deployment, the float completed a total of 183 profiles, the majority of which were acquired from the surface down to 600- or 1000-meter depth.

**1.2 Environmental data**

The float was equipped with several sensors to quantify seawater properties along the water column. The float was instrumented with pressure (DRUCK_2900PSIA, SN: 11587115), temperature and salinity sensors (SBE41CP_V7.2.5, SN: 13100) to measure the hydrological water properties. The biogeochemical properties were measured with oxygen (AANDERAA_OPTODE_4330, SN: 3489), fluorescence and backscattering (700 nm) sensors (ECO_FLBB_2K, SN: 6310). All the BGC Argo data that were used in this work, are made freely available by the International Argo Program (https://fleetmonitoring.euro-argo.eu/float/6903095). Before analysis, some derived variables were calculated based on a Python implementation of the Gibbs SeaWater (GSW) Oceanographic Toolbox of TEOS-10 (http://www.teos-10.org/). Firstly, Absolute Salinity (SA) was computed from Practical Salinity (PSAL) and pressure (dbar) measurements and Conservative Temperature (CT, °C) was computed from in-situ Temperature (T, °C). These two variables were then used to calculate the Potential Density (kg.m$^{-3}$) with a reference pressure of 0 dbar (Roquet et al., 2015). Regarding the oxygen parameter, the Apparent Oxygen Utilization (AOU, µmol.kg$^{-3}$) was computed as the difference between the oxygen concentration expected at equilibrium with air and the in-situ oxygen concentrations. The expected oxygen concentration was computed from solubility coefficients derived from the data of Benson and Krause (1984), as fitted by Garcia and Gordon (1992). Finally, the mixed layer depth (MLD), which is the upper part of the water column where salinity, temperature, and density remain vertically constant, was detected following De Boyer Montégut (2004). A reference value for density was taken at 5 dbar and the water column was considered to be mixed until the depth at which density deviates from this reference by more than 0.03 kg.m$^{-3}$

**1.2 Living and non-living particle data**

**1.1.1 Small phytoplanktonic and bulk particles**

To estimate chlorophyll-a concentrations emitted by phytoplanktonic cells, the float was equipped with a fluorescence sensor (ECO_FLBB_2K, SN: 6310). The backscattering (700 nm) sensor (ECO_FLBB_2K, SN: 6310) was used to quantify the amount of all particles (including detritus and phytoplankton). The backscattering coefficient signal (*bbp*) was decomposed to extract the signal of small particles (*bbsr*) from the raw signal which also contained spikes triggered by rare large aggregates passing in the flow field. This was done by applying the method proposed by Briggs et al., 2020 (see supplementary materials)

that decomposes the backscattering signal into its baseline (as a proxy of small particles, *bbsr*) and intermittent pulses (as proxy of larger aggregates). The *bbsr* signal represented the distribution of particles between 5 to 20 μm.

### 1.2.2 Underwater Vision Profiler 6 (UVP6-LP)

The float was also equipped with an in-situ camera, the Underwater Vision Profiler 6 Low-Power (UVP6-LP, SN: 000101) specifically designed to be deployed on autonomous platforms. The UVP6 detects and measures the size (from 0.102 to 16.40 mm in Equivalent Spherical Diameter, ESD) of zooplankton and various organic/biogenic matter such as marine snow and fecal pellets. Thanks to the float recovery, images of objects >500 μm could be retrieved as well and therefore, taxonomic identification of macrozooplankton and large particle classification could be conducted on the Ecotaxa platform (http://ecotaxa.obs-vlfr.fr). Taxonomic classification was initially assisted by a CNN (Convolutional Neural Network) algorithm to extract descriptive features from the images. From these, the taxonomic group of each image was predicted using a Random Forest algorithm. Then, living organisms' predictions have all been manually validated by taxonomic experts. All the images used during this work can be found here: https://ecotaxa.obs-vlfr.fr/prj/8801. A more detailed description of the UVP6-LP can be found in Picheral et al., 2022.

### 1.2.3 A broad size classification of particles (MiP and MaP)

To conduct a community analysis of marine snow and examine their spatio-temporal distribution, we applied a rough size classification on raw size- spectra data (>100 μm) provided by the UVP6 (without any plankton identification). In this case the assumption was made that zooplankton represents only a small fraction of objects sampled by the UVP6 (14.5% of images) compared to particles (85.5%). Hence, we consider its contribution as neglectable for this purpose. Then, all size-spectra were divided into two sub-classes: MiP (Micrometric Particles) and MaP (Macroscopic Particles). MiP concentrations were obtained by integrating the concentrations overall size classes between 0.1- and 0.5-mm. MaP concentrations were computed exactly in the same way but considering all size classes between 0.5- and 16-mm. The 0.5 mm threshold was used as detrital aggregates > 0.5 mm are considered marine snow (i.e., aggregates, Alldredge and Gotschalk, 1988) and was chosen for visualization purposes. However, the count/size measurements of the objects seen by the UVP6 contain zooplanktons and it is not possible, from the images, to determine whether an object smaller than 0.5 mm is a living organism or detritus due to the limited resolution. As a result, the MiP and MaP signals could be influenced by zooplankton abundances. However, the proportion of living organisms to the total particle counts (ESD > 500μm) are usually smaller than 10% (Stemmann and Boss, 2012).

Organic carbon content of MiPs and MaPs was calculated assuming that particle mass can be estimated using an empirically derived relationship for marine aggregates (Alldredge, 1998; Kriest, 2002 reference 2a of Table 1). Assuming a carbon:nitrogen ratio of 106:16, this yields an expression for the carbon content of a particle in a given size class characterized by its diameter ESD (in cm). Moreover, this formula is dedicated to estimate POC content of miscellaneous marine snow, including components such as fecal pellets. Multiplying the carbon content with the particle number in this size class (in

particles.m$^{-3}$), and integrating over the MiP and MaP size classes, respectively, we obtain the total POC content (mgC.m$^{-3}$) for
MiPs and MaPs.
Our study suggests the presence of vertical velocities within the water column which could significantly influence the marine
snow distribution (MiPs and MaPs) at depth. The literature provides various methods for estimating POC fluxes from imaging
devices. However, these approaches are based on particle size, sinking speed, and carbon content relationships which do not
include the influence of physical processes such as water masses vertical movement. Therefore, a simple POC fluxes estimates
from particle size would not accurately represent the actual fluxes in our study area. Furthermore, we couldn't disentangle the
contributions of particle sinking speeds from the vertical displacement of the water mass. Given these limitations, we chose
not to provide flux estimates.
The raw data corresponding to this section can be found on the Ecopart platform (https://ecopart.obs-vlfr.fr, project:
uvp6_sn000101lp_2021_WMO6903095_recovery).

### 1.2.4 Unsupervised morphological classification of marine snow

To better characterize the marine snow dynamics, an unsupervised classification method was used, following the previous
study on particle dynamics in the Arctic Ocean (Trudnowska et al., 2021). This provides information on particle size, shape,
gray level, and heterogeneity. This method allows one to classify rapidly and efficiently particle images in objective categories.
In summary, firstly, zooplankton and detrital particle images were separated, by manual validation and treated independently.
After this separation, a PCA was performed on morphological traits of the detrital particle images. Those specific traits
represented their size (e.g., area, perimeter), shade intensity (e.g., mean/median gray level), shape (e.g., symmetry, elongation),
and structure (i.e., homogeneity or heterogeneity, mostly based on the variability in gray level). To obtain a normal distribution
for each variable, extreme values (in that they were below or above the first and 95th percentile) were removed and each
variable was transformed by the Yeo-Johnson transformation (Yeo, 2000). This PCA led to the creation of a morphospace in
which each particle image can be located based on their morphological features. Then, a K-means clustering was performed
on this new morphospace using the first five principal components. The number of clusters, "k", was set to four. This specific
number, after several simulations, was chosen because it conducted the best partitioning with four clearly different groups of
particle morphology.
Finally, the concentration (nb.m$^{-3}$) of each morphotype was computed by multiplying the number of particles found in each
cluster and depth bin by the volume sampled by the UVP6. To study their spatio-temporal distributions, group concentrations
were interpolated according to depth and time with a resolution of 5 meters and one day respectively.
Since some groups in the morphospace partially overlap (Fig. 5.B), potentially affecting the quality of the classification, a
further selection process was implemented. This involved calculating the Euclidean distance between individual particle
images and their respective cluster centers. Subsequently, for each group, the first quartile of the Euclidean distance distribution

was computed, and only individuals with a distance smaller than the first quartile were included in the selection. The concentrations of the different morphotypes shown in figure 6 correspond to those of these 'exclusive members'. Objects with a too large distance (out of the first quartile) were not included in the further analysis.

**1.2.5 Selection of profiles corresponding to deep massive exports**

To characterize in an objective way significant changes in the deep marine snow spatio-temporal distribution, the Sequential T-test Analysis of Regime Shifts (STARS) method was used (Rodionov, 2004). This method, based on the Student's t-test2, analyses a dataset in a sequential manner. It compares each new observation with all the entries in the current 'regime' or group. If an entry significantly deviates from the average of that regime, it is marked as a potential 'shift point'. The algorithm then tests whether this detected shift is persistent over time. To characterize significant changes in the water column, marine snow concentrations were integrated, for each UVP6 profile, between 150 and 600 meters. Then, profiles detected in each significant 'shift' were selected and considered as belonging to (or 'inside') those features in marine snow distribution (see Fig. S1). Afterward, to compare the aggregates distribution outside and inside each feature, profiles one month before and one month after each feature were selected and merged, providing the mean of particle concentrations 'outside' each feature. The same protocol was applied to study the morphotype proportions according to depth. The latter were computed in four depth layers, between 0-100, 100-300, 300-600 and 600-1000 meters to see the evolution of the morphotypes relative abundance overall the water column.

**1.2.6 Eddy subduction detection**

To compare our findings with the literature, we applied the algorithm proposed by Llort et al., 2018 to detect ESP events from AOU and spiciness anomalies derived from BGC-Argo float data. Spiciness was defined as in Flament, 2002 and allows differentiating water masses with distinct thermohaline properties but with similar density. Then, we compared each profile's 3-bin smoothed values to a 20-bin average to detect anomalies for both AOU and spiciness. An event was classified as subduction-driven only if AOU and spiciness anomalies occurred at the same depth, which helped differentiate subduction effects from horizontal mixing. Thresholds of AOU anomaly < -8 µmol/kg and spiciness < -0.05 were set to detect these kinds of events. Finally, we compared the ESP occurrence with the export features spatial distribution detected with the STARS method. We also considered calculating vertical velocities according to Siegelman et al. (2020), but the temporal resolution of our data was not sufficient to enable this (pers. communication L. Siegelman).

**1.3 Satellite data and the TOEddies algorithm**

To identify mesoscale eddies, we implemented the TOEddies algorithm (The Ocean Eddy Detection and Tracking Algorithm, Laxenaire et al., 2024). TOEddies is based on the identification of mesoscale eddies as closed contours of Sea Surface Height (SSH) surrounding an extremum. This relies on the principle that in a geostrophic balance, SSH isolines align with current streamlines. As a result, a maximum (minimum) SSH that is surrounded by a closed circulation is classified as an anticyclonic

(cyclonic) eddy. To track the eddies over time, the algorithm takes advantage of the fact that daily eddy displacements are
small compared to their dimensions, resulting in overlapping areas between successive days. This methodology allows the
derivation of trajectories and the detection of events where eddies merge or split, providing a mechanism for tracing the origin
of sampled water masses which is a critical component for the objectives of this study .
For the purposes of this study, the TOEddies algorithm was applied to the Absolute Dynamic Topography (ADT, a proxy for
SSH) and its associated with geostrophic currents. These 1⁄4 gridded daily maps are produced by Ssalto/Duacs and distributed
by the Copernicus Marine Environment Monitoring Service (http://marine.copernicus.eu/) in the version released in April 2018
(DT18; Pujol et al., 2016; Ballarotta et al., 2019). Each identified eddy was characterized by two radii associated to the two
eddy boundaries defined in TOEddies: the outermost contour ($R_{out}$), and the contour with the maximum averaged geostrophic
velocities ($R_{Vmax}$ and associated velocity $V_{max}$).
Eddies identified by the algorithm were collocated with float profiles (Chaigneau et al., 2011; Laxenaire et al., 2019; 2020).
This allows us to categorize the profiles according to whether they sampled a cyclone, an anticyclone, a region outside of the
influence of mesoscale eddies, or an area at the interface of two eddies. The application of TOEddies to ADT maps, coupled
with the collocation of detected eddies with Argo profiles, has proven to be a successful combination for exploring eddy
dynamics in the southeast Atlantic (Laxenaire et al., 2019, 2020; Ioannou et al.,2022; Baudena et al., 2023).

## 1.4 Lagrangian diagnostics

Several Lagrangian diagnostics have been computed at the location of each profile using satellite-derived currents and
environmental variables. First, for each station (i.e., profile location), a region was defined as representative of the water parcel
sampled by the float. In this study, this region was a circular neighborhood with a radius r of 0.1° around each profile location.
This distance allowed us to smooth the satellite uncertainties and has been used in previous studies (Chambault et al., 2019;
Baudena et al., 2021; Ser-Giacomi et al., 2021; Fabri-Ruiz et al., 2023). This circular shape was then filled with virtual particles
(nearly 300) separated by 0.01°. Then, for each virtual particle, a given Lagrangian diagnostic was computed (detailed below).
This resulted in about 300 values for each sampling station. These have been averaged, leading to a value of a given Lagrangian
diagnostic for each profile. The velocity field we used have been derived from both altimetry and delayed-time model
assimilation data and includes both the geostrophic and the Ekman components (Copernicus CMEMS product MULTIOBS
GLO PHY REP 015 004-TDS). It has been used backward in time to advect each virtual particle (within the representative
water parcel) from the profile day until an advective time ($\tau$). Different $\tau$ values were used, ranging from 5 to 45 days. Hence,
for each profile, each diagnostic has been calculated using different advective times. The first diagnostic we implemented was
the Finite-Time Lyapunov Exponent (FTLE, days-1; Shadden et al., 2005). By construction, intense FTLE values (typically
disposed in filament shapes) are found at the edge between two water masses that have been widely separated in the preceding
days. As these two water masses come from distant locations, they are likely to share different hydrological characteristics,
such as temperature, primary production or biological activity (d'Ovidio et al., 2010; Haller, 2015; Lehahn et al., 2018). For
these reasons, FTLE is useful to identify oceanic fronts and can be considered as proxies for water masses convergence and

thus, possibly, associated vertical downwelling (McWilliams, 2016). FTLEs were computed as in Shadden et al., (2005), with an initial separation of 0.1°. We implemented as a second diagnostic the Lagrangian chlorophyll-a (mg.m$^{-3}$), which is the mean chlorophyll content calculated along the backward virtual particle trajectory. This diagnostic determines whether the seawater parcel sampled at a profile location was rich in chlorophyll in the preceding days. For the chlorophyll, we used the delayed-time satellite product "OCEANCOLOUR GLO BGC L4 MY 009 104-TDS" provided by CMEMS Copernicus website (0.25° resolution, provided daily).

## 2 Results

### 2.1 Circulation and water mass spatio-temporal distribution

The lateral variability in thermohaline properties (Fig. 2) is driven by mesoscale eddies and submesoscale features, which carry distinct signatures and interact dynamically, creating spatio-temporal heterogeneity in water masses. Visual inspection of the TOEddies algorithm results, coupled with analysis of hydrological data provided important information about the history and origin of the water masses sampled by the float. In the following, we separate the life history of the float trajectory in different periods. During April to October 2021 (fall and winter), the float was trapped within a cyclonic eddy shed from the southern Benguela upwelling front (Baudena et al., 2023, preprint). During this period, temperature, salinity, and density were relatively stable. Figure 2 shows that the MLD varied between near-surface depths and 200 m. Analysis of the AOU time series shows predominantly weak positive or even negative values only above the MLD (Fig. 2.F). Regarding the FTLE time series, values remained low compared to the entire time series (less than 0.25 days$^{-1}$) except around August 2021 when the targeted eddy merged with another Benguela Upwelling cyclone (0.45 days$^{-1}$). Overall, the float remained near the center of the cyclonic eddy, which explains the low FTLE values. After the merging, the float continued to move within the new cyclonic structure. This cyclonic eddy merged again (in early October 2021) with an Agulhas cyclone that formed along the Agulhas Bank, the southern shelf-edge of Africa (Penven et al., 2001; Lutjeharms et al., 2003). This type of cyclone originates from the southern African continental slope from barotropic instabilities and enters the ocean interior at the northern edge of the Agulhas Current Retroflection (Duncombe Rae, 1991) with very different hydrographic characteristics (warmer and saltier water masses, Giulivi and Gordon, 2006) compared with the Benguela upwelling cyclones.

After exiting the cyclone, the float remained within its periphery. In mid-October, the float was at the interface between the cyclone and an anticyclonic eddy. Then, the float moved southward until May 2022. The water column was more stratified and characterized by a shallower MLD (typical for spring and summer). During this period, two phases of the life history of the float can be identified. During the first phase, warmer and saltier waters were observed between 200 and 800 m depth. These waters clearly originate from Agulhas Current water masses, very likely originating from an injection in the region of the Agulhas rings generated at the Agulhas Current Retroflection (Laxenaire et al., 2018). The second phase is characterized by colder and fresher water masses. Indeed, between January and March 2022, the float crossed the Subantarctic front and

continued to move southward along the edge of an Agulhas ring, a relatively old Agulhas cyclone, as well as a Subantarctic Front cyclone. Finally, in April 2022, the float passed next to another Subantarctic Front cyclone, which could explain the colder and less salty water masses during this period. Moreover, specific patterns in the AOU and the density time series are also observed during the transition between these two phases. Indeed, the AOU time series shows smaller values (close to 0 $\mu mol.m^{-3}$) at depths between 200 and 1000 m. The latter are also associated with important horizontal isopycnal variations (steep slopes), reaching depths close to 900 m and coinciding with high FTLE values (above 0.45 $days^{-1}$).

The final period spans May to September 2022 and is marked by three high-intensity mesoscale features that influence the water column down to about 800 to 1000 m depth. These features are characterized by a deeper MLD and the presence of much warmer and saltier water masses all over the water column. The AOU time series shows a strong signal during this period, with negative values observed at depths around 800 m. This period was also characterized by intense FTLE values throughout. The TOEddies analysis showed that the float profiled different cyclones and anticyclones during this period.

In summary, during its deployment, the float sampled water masses with significantly different hydrological characteristics and from multiple geographical zones. At this stage, one of the main results to keep in mind is that the float sampled interface zones between anticyclonic and cyclonic structure. Overall, these observations highlight the dynamic nature of the study area, characterized by contrasting water mass properties and their temporal variations throughout the year.

## 2.2 Spatio-temporal distribution of marine snow during the survey

From April to October 2021 (Fall and Winter), surface chlorophyll-a concentrations and the *bbsr* coefficient remained relatively low (Fig. 2.E.F). Afterwards, a significant change occurred between the end of October 2021 and May 2022 (Spring and Summer). This period coincided with the seasonal production of phytoplankton, as evident in the chlorophyll-a concentration time series. Also, a noticeable increase in the *bbsr* coefficient indicates greater concentrations of small particles in the surface (Fig. 2.E). Moving forward to the period between May and September 2022, there was a noticeable decline in chlorophyll-a concentration. Consequently, weaker concentrations were observed at the surface, and the *bbsr* coefficient was also reduced. In general, the spatio-temporal dynamics of MiP and MaP exhibit notable differences (Fig. 3). MiPs were generally more abundant than MaPs, ranging between 1.5- and 7.5 $part.L^{-1}$, while MaP concentrations ranged from 0 to 1.6 $part.L^{-1}$ (Fig. 3). Both types of particles displayed higher mean concentrations above the MLD than below. MiP were consistently present throughout the float deployment, with a slightly higher abundance observed between October 2021 and May 2022. Conversely, the distribution pattern of MaPs differed. Their concentration was relatively low from the beginning of the time series until October 2021. Subsequently, their surface abundance significantly increased during the phytoplankton production period between the end of October 2021 and May 2022. Overall, these observations highlight the distinct spatio-temporal dynamics of MiP and MaP. A key result in the marine snow spatial distribution is that three different "columns" of increased particle concentration (in particular MaPs) were observed between the surface down to 600 meters. These features

lasted for approximately a month. The first one took place between October 8 and 23, 2021, the second one occurred from
November 25 to December 22, 2021, and finally, the last one spanned approximately between March 3 and April 28, 2022.

**2.3 Spatio-temporal distribution of MaP morphological groups during the survey**

The unsupervised clustering method led to the identification of four morphotypes distinguished by their size, circularity,
brightness, and homogeneity (Fig. 5.A.B). The first morphotype was characterized by small, dark, and predominantly circular
particles. The second morphotype comprised elongated objects with varying degrees of brightness. The third morphotype
consisted of bright, fluffy, and diverse marine snow particles. The fourth morphotype encompassed larger marine snow
particles, often with the form of aggregated structures with some heterogeneity. All the identified morphotypes were
predominantly found in the surface layer. Specifically, the elongated particles (Fig. 6.B) exhibited a distinct concentration
increase from late September 2021, maintaining a stable abundance until early May 2022. Before and after this period, their
presence was considerably smaller. Additionally, these elongated particles displayed a strong positive correlation with
chlorophyll-a concentration in the surface layer (0-100m), as evidenced by a significant Spearman correlation coefficient of
0.65 (p-value = $9.3.10^{-22}$, Fig. S3). In contrast, the other three morphotypes (small, bright, and aggregates, Fig. 6.A.C.D)
exhibited different dynamics. They were present in all three MaP features described above, whereas the elongated particles
did not appear to be extensively involved. The dense aggregates morphotype (purple in Fig. 5.C) is the most abundant during
these three features (Fig. 6.A-C).

**2.4 Vertical distribution of particle community composition inside and outside massive export features**

Here we analyze the distribution of the four morphotypes in the water column during the three export features (Fig. 5.D).
Indeed, a shift is observed in their relative proportion between the surface and the layers below. Specifically, the abundance
of small particles (salmon pink in Fig. 5.D) and dense aggregates (purple) increases, while the proportion of elongated and
bright morphotypes decreases. For example, in column 3, the proportion of small morphotypes incremented from 12% to 23%
and from 25% to 47% in dense aggregates. Furthermore, the distribution of morphotypes within each export feature did not
change considerably, except perhaps in the case of column 2.

**2.5 The eddy dynamic context**

The first and the second export feature occurred at the boundary between a cyclonic and an anticyclonic eddy (Fig. 1. B.C).
This region was therefore characterized by very intense submesoscale frontal dynamics. During the second feature (Fig. 1. C),
the position of the cyclone did not change significantly. The float entered the cyclone but remained close to its $R_{Vmax}$ boundary
and thus within the submesoscale frontal region of the cyclone edge. Indeed, the eddy frontal regions extend over an area of
10-20 km across the region of eddy maximum azimuthal velocity (i.e., across $R_{Vmax}$) as shown in Barabinot et al., (2023). This
second feature ended when the float was expelled from the cyclone and began to be advected southward, leaving the frontal

region. The third export occurred when the float was advected around the southern edge of a large anticyclone (Fig. 1.D), again a submesoscale frontal region.

**2.6 Impact of the three features on the upper 600m water column**

Figure 4 clearly illustrates the high abundance of particulate carbon associated with the three export events, with their influence extending down to 600 m depth. The abundance increased by a factor of 2-3 (with a factor of more than 7 during the first feature) compared to periods when the Argo float was moving 'outside' these frontal regions. An average increase in MiP POC of about 25% is also observed for the first and the second features. Importantly, the three features occurred during the surface production period and do not show a signature in smaller particle abundances (as indicated by the absence of such a pattern in fluorescence and *bbsr*, Fig. 2.D.E). In particular, MiP and MaP abundances between 100 and 1000 m were significantly correlated with intense FTLE values (Spearman's rank correlation coefficient of 0.4, p-value = $4.10^{-4}$ and 0.49, p-value = $1.3.10^{-5}$ respectively, Fig. S4-S5). Mean MiP abundance, at this depth, was also significantly correlated with low AOU values (coefficient correlation of -0.59, p-value = $1.6.10^{-8}$, Fig. S6).

**3 Discussion**

**3.1 Drivers of marine snow production in the surface layer**

The analysis of the chlorophyll-a time series helped to characterize the seasonality of primary production in our study area. In particular, from the end of October 2021 to May 2022 (spanning spring and summer), there was a significant increase in surface layer chlorophyll-a concentrations (Fig. 2.D). This observation is consistent with the existing knowledge on the seasonality of phytoplankton in the southern part of the Atlantic Ocean (Thomalla et al., 2011) and led us to consider this period as the productive period.

Regarding the spatiotemporal distribution of MiP and MaP (Fig. 3.B), the abundance of MaP (i.e. aggregates) in the surface layer (0-100m) was significantly correlated with the surface Lagrangian chlorophyll-a 15 days backward (see Fig. S2) and high FTLE, indicating a strong biophysical coupling via several synergistic mechanisms (enhanced primary production, coagulation, transport). Ocean eddies, especially cyclones, have the potential to enhance primary production by upwelling nutrient-rich water from deeper layers to the surface (Benitez-Nelson et al., 2007; Ascani et al., 2013; Cornec et al., 2021). This leads to the biomass accumulation at the edges of cyclones due to water mass divergence (water mass flow from the center to the cyclone periphery), resulting in aggregate formation between submesoscale structures (Lima, 2002). Eddies can also move phytoplankton patches through eddy stirring or trapping (McGillicuddy, 2016), leading to the accumulation of organic matter at the eddy boundaries (Fig. 7.1). This implies that actively growing phytoplankton transported by the water body in the previous days could have contributed to increase the abundance of MaPs by aggregation of organic matter and biological activity. Among the MaP, large particles classified as elongated type were mostly found at the surface and they were the best correlated with surface chlorophyll-a. Their origin is difficult to determine. Their filamentous morphology suggests a

resemblance to chain-forming diatom colonies. The average size of particles classified under the 'elongated' morphotype is approximately 820 µm (±96 µm). Given their size and the study area, it's likely that some of these particles could include colonies of *Chaetoceros sp.*, which are known to form filamentous structures and to dominate phytoplanktonic blooms during the austral summer (Laubscher et al., 1993). They could also be Euphausiid fecal pellets, however, this kind of organisms was not observed at the surface by the UVP6 which could be attributed to their potential avoidance behavior, making their identification challenging. However, given their size (>500 µm), it is unlikely that they are copepod fecal pellets, most of which are smaller than 200 µm (Møller et al., 2011). The other categories of MaPs are also abundant and could be the result of the aggregation into larger particles by biophysical coagulation or trophic activities. Coagulation is responsible for the production of large particles when both particle concentrations and stickiness are high. Additionally, a heterogeneous particle size distribution enhances coagulation, as smaller particles have a higher probability of colliding with larger particles (Hunt, 1982; McCave, 1984; Jackson, 1990). Through this mechanism, particle encounter rates are enhanced by Brownian motion, shear and/or differential settling resulting in the formation of a single, larger particle (Jackson, 1990; Stemmann et al., 2004) on a timescale of a few days, as already suggested in the Southern Ocean (Jouandet et al., 2014). By affecting photosynthesis and water movement, the observed intense mesoscale and submesoscale dynamics, (Fig. 1.B.C.D), could significantly affect particle concentration and the probability of aggregation. The second aggregation mechanism by zooplankton grazing, sloppy feeding (Lampitt et al., 1990) and fecal pellet production (Turner, 2015), occurs at long temporal scales due to the slower growth time of zooplankton. Our observation strategy with a float cannot rule this aggregation mechanism out. However, it seems that typical mesozooplankton fecal pellets (a few hundred µm) were not observed in abundance in, at least, three of the four morphotypes. We haven't also observed a clear increase in zooplankton abundance during the three particle distribution events. However, a slight increase in copepod abundance was noted during the first and third features within the first 100 meters (see Fig. S7). In our case, it is more likely that physical coagulation had a greater influence on aggregate formation, however, we cannot rule out trophic biological aggregation. In fact, MaP were present at the surface only during the period of maximum chlorophyll-a concentration and were associated with intense FTLE values (greater than 0.45 days$^{-1}$), which can be used as a proxy of water masses convergence (i.e. frontal zones as in Prants, 2013; Hernández-Carrasco et al., 2018). In other words, MaP concentrations may have been enhanced due to the combination of two important factors for coagulation according to the model of Jackson (1990), increased shear associated with high hydrodynamics at front and enhanced phytoplankton biomass.

**3.2 Coupling between massive export of large aggregates and regional ocean dynamics**

Analysis of the spatio-temporal distribution of particles has revealed three regions of enhanced concentration of particles at depth, suggesting that vertical export of organic matter occurred in these areas during the productive season. These were characterized by intense MaP and MiP abundances extending vertically from the surface to depths of about 600 m. These export regions were associated with high FTLE values suggesting frontogenesis-driven mechanisms during which there could be a strong coupling between gravitational sinking of particles and intensified physical vertical velocities.

### 3.2.1 A submesoscale frontogenesis-driven mechanism

The abundance of particles such as MaP and MiP below the MLD (100-600m) was positively correlated with the surface Lagrangian FTLE as well as for the small, bright and aggregated morphotypes. Large FTLE values can identify convergence of water masses with different hydrological properties, creating intense density gradients. It is known that intense velocity and density gradients arise in correspondence with mesoscale (geostrophic) and submesoscale (ageostrophic) frontogenesis, e.g. at the edge of mesoscale features (Freilich and Mahadevan, 2021). At such fronts upper-ocean waters converge and subduct beneath the mixed layer into the stratified pycnocline (e.g. Omand et al. 2015) or also in deeper layers (e.g. Llort et al., 2018). These regions are typically associated with significant vertical velocities (up to 20 m/day). Frontal patterns, in the potential density time series, are clearly seen extending into the ocean interior (Fig. 2.C) and are also found to correspond to the three intense export features. During these features, the signal reached about 1000 m. All of them were located at the periphery of mesoscale eddies (Fig. 1. B.C.D), where the vertical velocity is expected to be higher (Thomas et al., 2008; Freilich and Mahadevan, 2019). Unfortunately, it was not possible to calculate vertical velocities from the data provided by the BGC Argo profiler.

Previous studies in a similar mesoscale physical context have shown that mesoscale activity can result in the intrusion of small POM-rich water masses into the mesopelagic layers (Omand et al., 2015; Llort et al., 2018). In our study, the three features described, occurred from the MLD to a depth of about 600 m or deeper (i.e. over a large part of the water column), whereas the subduction described in Llort et al. (2018) occurred well below the MLD in the mesopelagic in depth layers thickness of about 50-100 m. We applied the Llort et al. (2018) algorithm to identify ESP events across all profiles. Only 28 profiles were classified as ESP events (see Figure 1.A, yellow dots). Of these, seven were located near export features described in this study. However, increases in *bbsr*, chlorophyll-a, MiP, and MaP were not consistently observed across these ESP events. Furthermore, the ESP events were generally less than 100 meters thick, whereas the export features we identified extended through much of the water column. These findings suggest that while some export features may be linked to ESP events, the ESP alone does not appear to be a fundamental trigger for these features. This suggests that we are describing a slightly different mechanism compared to Omand et al. (2015) and Llort et al. (2018) responsible for the accumulation of organic matter at depth, but also one related to mesoscale and submesoscale processes. This highlights the fact that the influence of these physical mechanisms is a challenging area of research.

### 3.2.2 The key role of the gravitational carbon pump and the MLD

Historically, particle size has been considered as the primary factor influencing settling velocity (Guidi et al., 2008). However, with advances in imaging systems, it has become increasingly clear that particle morphology has a significant effect on settling velocities (Iversen and Lampitt, 2020; Cael et al., 2021; Williams and Giering, 2022). This process supports the idea that larger (large ESD, size characteristic) and more compact (non-porous, morphological property) particles will sink faster. In this study, direct estimates of settling particle velocity (e.g. export plume methodology; Stemmann et al., 2002; Briggs et al., 2020;

Trudnowska et al., 2021) could not be computed due to the non-completely Lagrangian nature of the float. However, it is possible to assume that the morphotypes identified in this work have different sinking velocities with the highest for small/dense and dark aggregate morphotypes (which dominate in the deepest water column, Fig. 5.D) and the lowest for the elongated morphotypes because they are only found at the surface (Fig. 5.B). In addition, the particles with the higher velocities should be rapidly exported to depth, so they should spend less time in the upper water column than particles with the lowest sinking velocity. In other words, the latter will be more exposed to remineralization.

Furthermore, the analysis of the morphotype distribution during the three export features showed a shift in the group proportions between the surface layer (0-100m, which is approximately the mean depth of the MLD, all profiles included) and deeper layers (100-1000m). This shift favored small and dense aggregate groups (in terms of proportion) between these two layers. These morphotypes probably have the highest sinking speed (Soviadan et al., 2024). This supports the idea that the MLD could act as a physical barrier (Hosoda et al., 2010) and only sinking particles were able to cross it. This conclusion is also supported by the fact that, in most profiles, only MiP and especially MaP are located below the MLD, while small labile and refractory particles (*bbsr*) and chlorophyll-containing particles were rarely found below the MLD. Among the MaP, the sinking explanation is valid for three morphotypes (small particles, bright and dense aggregates), but not for the elongated types that we hypothesized to be chain-forming diatom colonies (*Chaetoceros sp.*). This result further emphasizes the critical importance of considering morphology and not just particle size when analyzing the size of particles in marine snow dynamics and when calculating vertical fluxes using only allometric relationships.

Another result is that features 1 and 3 show minimal changes in morphotype proportions between layer depths. In fact, it would be expected that the relative proportion of each morphotype would vary with depth due to their presumed different settling rates. It is possible that the morphotypes do not have different settling rates and settle at similar rates. Alternatively, there may be some other mechanism at work that is significantly more important than settling velocity and is responsible for transporting particles to deeper regions, which will be discussed below.

### 3.2.3 Possible coupling with vertical velocities

During the three described features, MiP abundance between 100 and 1000 meters was significantly correlated with low AOU values. AOU is a proxy for the age of water masses (Sarmiento, 2006), with lower AOU values indicating that the water masses had a more recent contact with the surface. The significant negative correlation between AOU and 100-1000 m MiP abundance suggests a physical downward transport leading to the formation of these features. In addition, we know from the eddy detection algorithm that each export feature was detected in the frontal region separating two adjacent eddies of opposite polarity. Mesoscale activity is known to play an important role in the vertical distribution of water masses.

In addition, surface frontal regions develop ageostrophic secondary circulations (Lévy et al., 2018) that lead to intense vertical velocities, which can be stronger at the edges of cyclonic structures and can be on the order of 100 m d$^{-1}$ or more (Mahadevan

& Archer, 2000; Mahadevan et al., 2016; McWilliams, 2016; Freilich & Mahadevan, 2019; Siegelman et al., 2020; Capó et al., 2021; Tarry et al., 2021). This is further supported by several studies that show that the presence of mesoscale eddies greatly enhances the vertical transport of water properties (Omand et al., 2015, Llort et al., 2018, Couespel et al., 2024). In the study region, mesoscale ocean dynamics are particularly active, with eddies generated by the Agulhas Current (the Agulhas rings), the Agulhas Bank (the Agulhas lee cyclones), the Benguela upwelling, and subantarctic front cyclones. All can influence the water column to depths of up to 1600 meters (Schmid et al., 2003). While these dynamics are known to exist, their impact on the vertical exchange of properties through the water column remains to be assessed. However, a recent study (Siegelman et al., 2020) has shown that such vertical velocities are enhanced below the MLD, which acts as a buffer for them, by relatively deep submesoscale fronts. This result may explain why only large particles (i.e., MiP, MaP, and morphotypes) involved in the gravitational pump and capable of crossing the physical barrier represented by the MLD are entrained in this process. However, we lack information on the 3D context of the water mass, both in terms of currents and biogeochemical properties, which makes it impossible to determine what happened at depth before the float passed. Consequently, it's impossible to rule out the action of lateral advection on the mechanism described in this study. Nevertheless, the strong correlation between the described events, the FTLE, and the positions of the eddies suggests a significant link to vertical rather than lateral advection. In addition, previous studies have shown that frontal structures identified by Lyapunov Exponents ridges at the surface can persist in the water column (Bettencourt et al., 2012), corroborating our findings. The hypothesis that the gravitational pump may be associated with physical processes related to mesoscale activity has already been proposed by Guidi et al. (2007). The authors found that the natural gravitational settling of particles combined with the associated vertical velocities of the water masses triggered the export of particles into the deep Atlantic. Such a physical mechanism associated with the gravitational pump would be a key process of carbon export to the deep ocean observed in our study.

**4 Conclusion**

As described in the conceptual schematic (Fig. 7), mesoscale frontal dynamics between cyclones and anticyclones seem to have a profound effect on particle distribution down to 600 m depth. We propose that particle concentration at these locations is enhanced by increased primary production and/or horizontal convergence induced by frontal activity or trapping. As a result, coagulation processes may have been enhanced, with the ultimate result being the formation of larger aggregates. These particles could have sunk below the MLD, in particular the large and/or dense ones, due to their higher sinking velocities. The consistently observed vertical distribution of particles from the surface to a depth of 600 m suggests that their downward transport is extremely rapid. We propose that such features result from the combination of the relatively large sinking speed of such particles (likely due to their great density) and the enhanced vertical motion of the water induced by submesoscale frontal dynamics, in particular below the MLD. If such a phenomenon is influenced by deeper mesoscale structures, it is likely to extend into deeper layers. Such vertical transport is an integral part of the physical particle injection pump, particularly a unique frontal subduction pump, characterized by intense vertical velocities that might materialize in intense frontal regions.

However, the relatively coarse temporal resolution of our dataset did not enable us to calculate the according vertical velocities,
and we hence could not disentangle the gravitational and subduction pump in more detail.
Our study suggests that small-scale ocean dynamics represent an efficient particle injection pump that can enhance the
efficiency of the biological pump by increasing the depth of carbon transport. It is crucial to note that the explicit mechanisms
underlying these features are not fully defined due to the temporal and spatial limitations imposed by the Argo float's lack of
resolution. Additionally, the submesoscale nature of these features introduces high variability in both space and time. This
study underscores the significance of implementing repeated sampling campaigns or the use of gliders equipped with UVP6
and further sensors (e.g. microstructure) and focusing on the interface zones between eddies. Such studies would help to
validate our proposed mechanism and to disentangle the relative contribution of the various PIPs processes (such as ESP). In
particular, focus on such studies needs to be put on resolving the water movements (e.g. the vertical velocities) to enable the
separation of the passive sinking and vertical translocation of particles. Since mesoscale and submesoscale structures are
ubiquitous in the ocean, it would also be interesting to quantify the influence of this type of process on the overall carbon
budget and to determine the extent to which it contributes to carbon sequestration in the deep ocean, information that is crucial
in the context of global change.
**5 Author contribution**
A.A., R.K.; SS and L.S.: designed the study
A.A.: conducted data analysis, interpreted results, drafted the manuscript, under guidance of RK and LS
AB and RL.: supported the study by environmental background data analysis, revised manuscript.
R.K. and S.S.: contributed to the conception of the study, revised manuscript.
R.K.: coordinated the recapture of the float
L.S.: contributed to the conception and design of the study, supervised the analysis, substantively revised the manuscript.
**6 Competing interests**
The authors declare that they have no conflict of interest.
**7 Acknowledgments**
This study would not have been possible without the support of the crews of the Research Vessel Sonne from Germany and
the SA Agulhas II from South Africa. We acknowledge the work of Marc Picheral, Edouard Laymarie, Antoine Poteau and
Camille Catalano from the Laboratory of Villefranche sur Mer in helping with the UVP6 logistic and quality control. We also
thank the Quantitative Imaging Platform of Villefranche-sur-Mer (PIQv) for providing us with their taxonomic expertise. This

work was supported by the TRIATLAS project, which has received funding from the European Union's Horizon 2020 research and innovation program under grant agreement No 817578. RK furthermore acknowledges support via a Make Our Planet Great Again grant from the French National Research Agency (ANR) within the Programme d'Investissements d'Avenir ANR-19-MPGA-0012 and funding from the Heisenberg Programme of the German Science Foundation KI 1387/5-1.

**Figures**

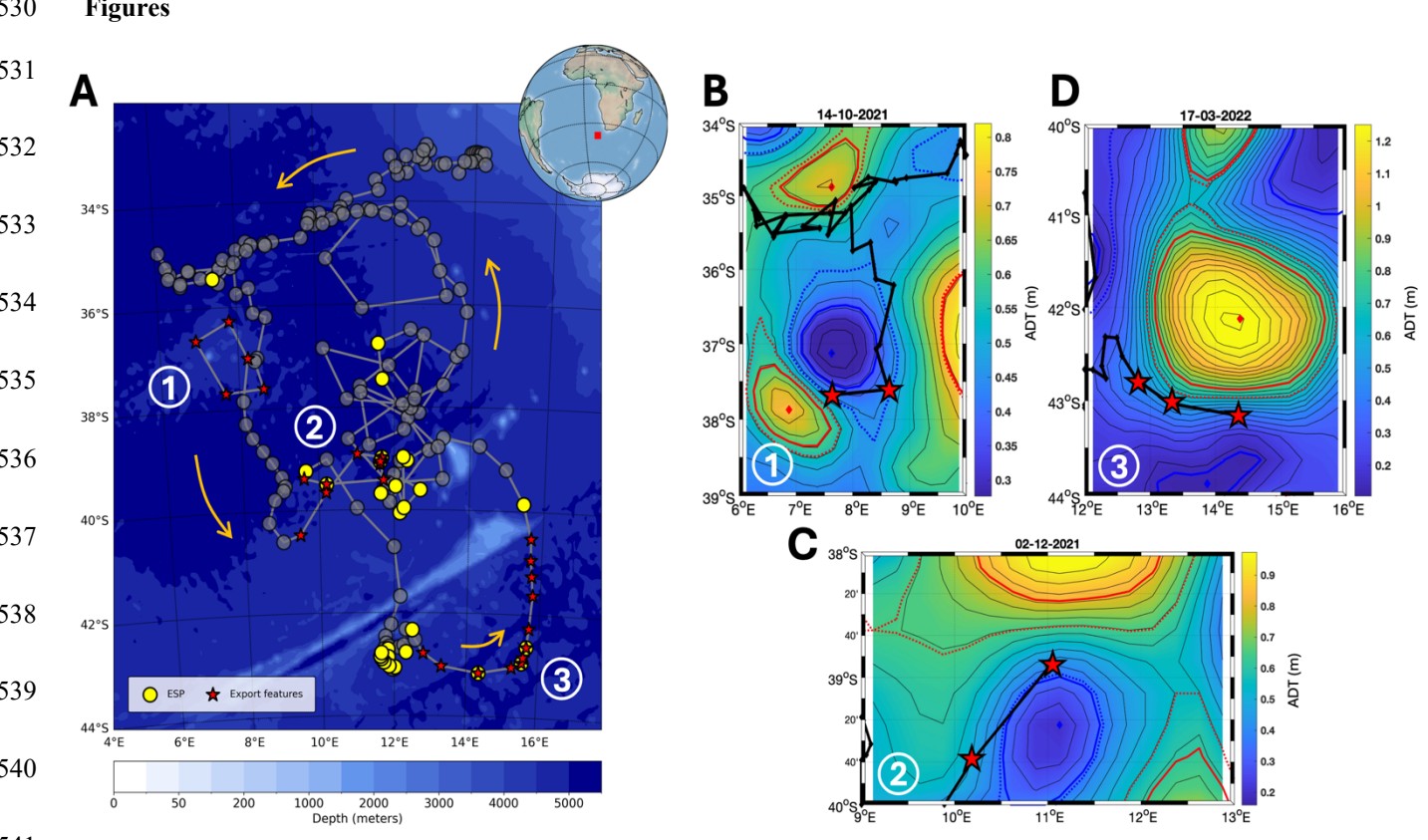

**Figure 1: (A) Float trajectory (thick gray line) during its entire deployment from April 2021 to September 2022 in the South-East Atlantic Ocean. Orange arrows indicate the direction of the float trajectory. Stars highlight export features described in our study (1 to 3 in panels B, C and D). Yellow dots indicate Eddy Subduction Pump (ESP) events (defined as in Llort et al., 2018). Shaded gray dots indicate profiles during which no export features were detected. The background map is the bathymetry of the study zone. (B), (C) and (D) panels show ADT field snapshots with the float trajectory (thick black line) during each export feature. Cyclones and anticyclones are associated with blue and red colors, respectively. Diamonds show eddies centroid. Thick solid lines correspond to eddies maximum speed. Thin solid lines correspond to ADT isolines. Dashed lines to eddies outer limit definition.**

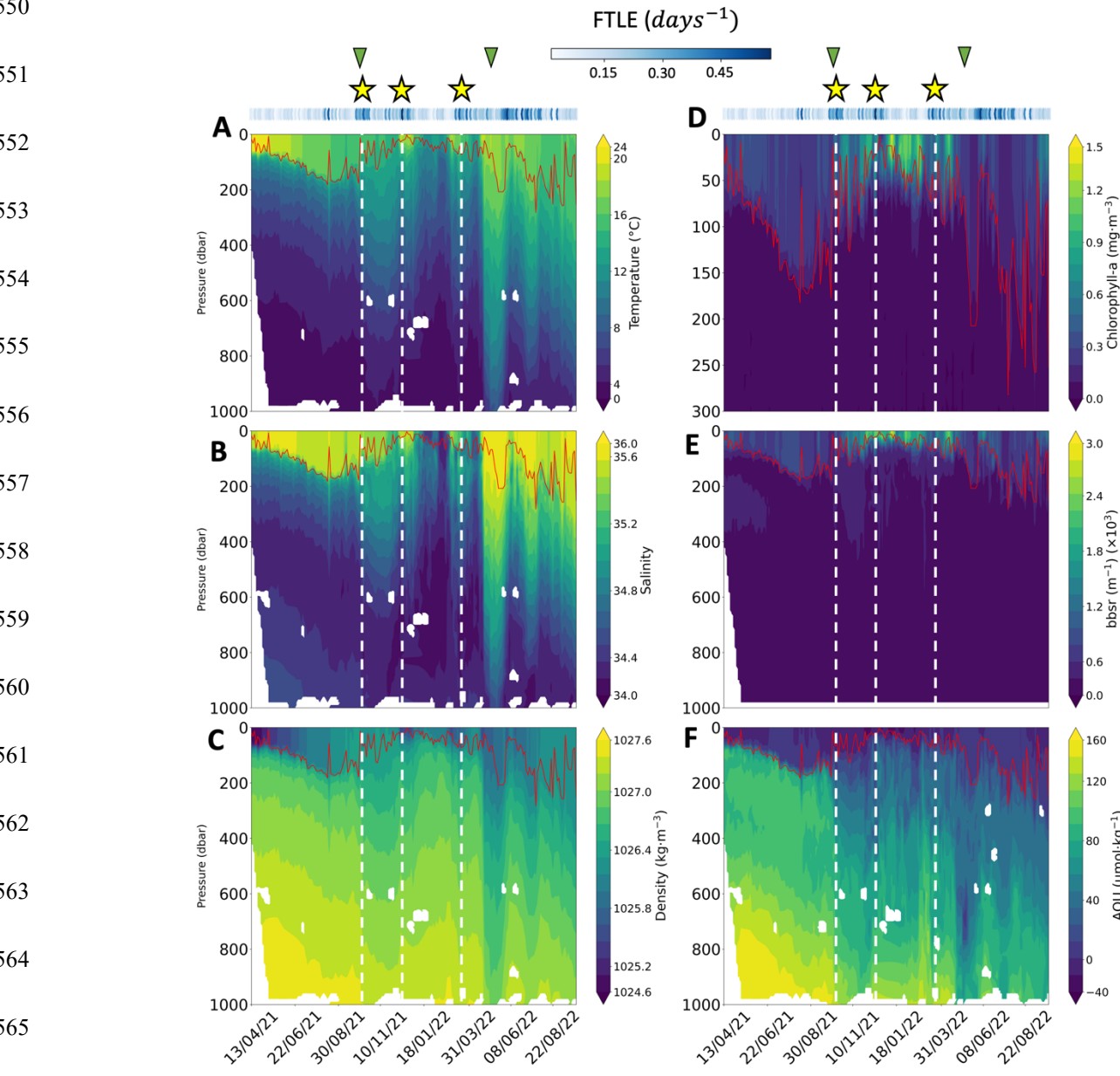

**Figure 2: Hydrographic and biogeochemical properties as a function of time and depth along the float trajectory (1.5 years, from April 2021 to September 2022). The panels show (A) temperature, (B) salinity, (C) potential density, (D) chlorophyll-a concentration, (E) log10 of 700 nm optical backscatter bbsr, (F) Apparent Oxygen Utilization (AOU). The red solid line in each panel shows the mixed layer depth (MLD). The white dashed lines and the yellow stars show the location of intense exports of particles. The green triangles indicate the start and the end of the production period. Blue dots on top of the panel show the mean five days backward FTLE (days$^{-1}$) for each profile location.**

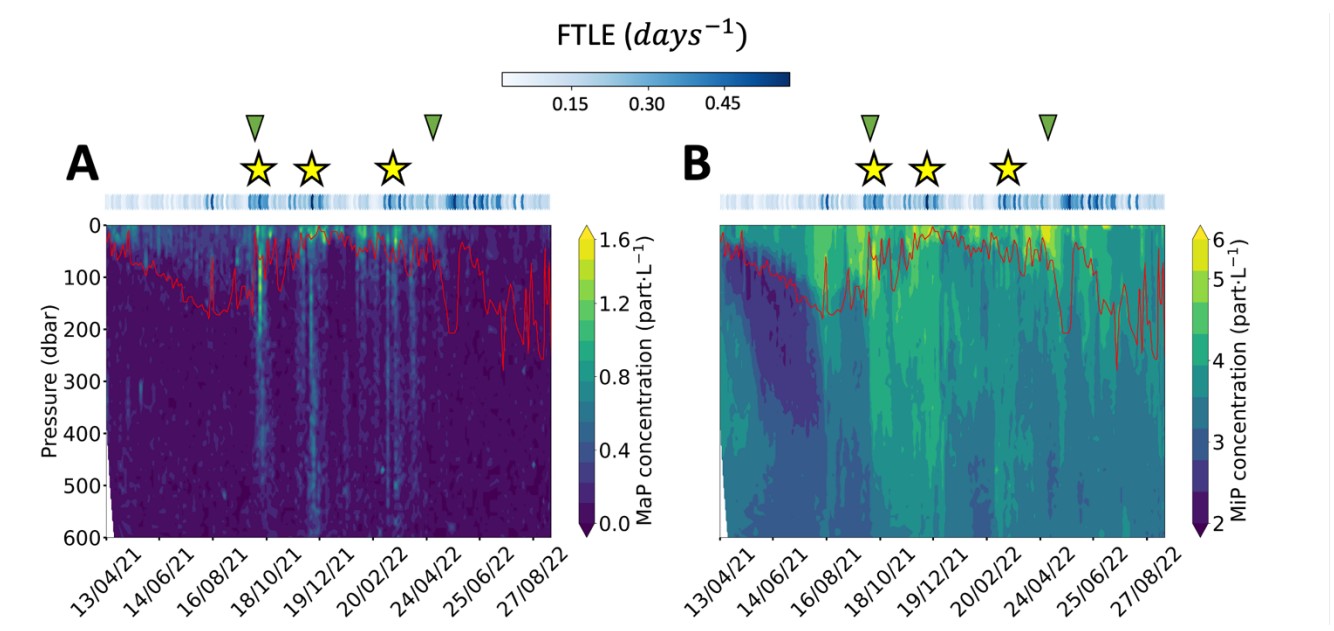

**Figure 3: Particles spatio-temporal distribution. The panels show, (A) MiP (Micrometric Particles 0.1 < ESD < 0.5 mm) concentrations, (B) MaP (Macrometric Particles 0.6 < ESD < 16 mm) concentrations. Both concentrations were log transformed. Red solid line represents the MLD. Yellow stars show the location of intense exports of particles. Green triangles give the start and the end of the production period. Blue dots on top of the panels show five days backward FTLE (days$^{-1}$) for each profile location.**



















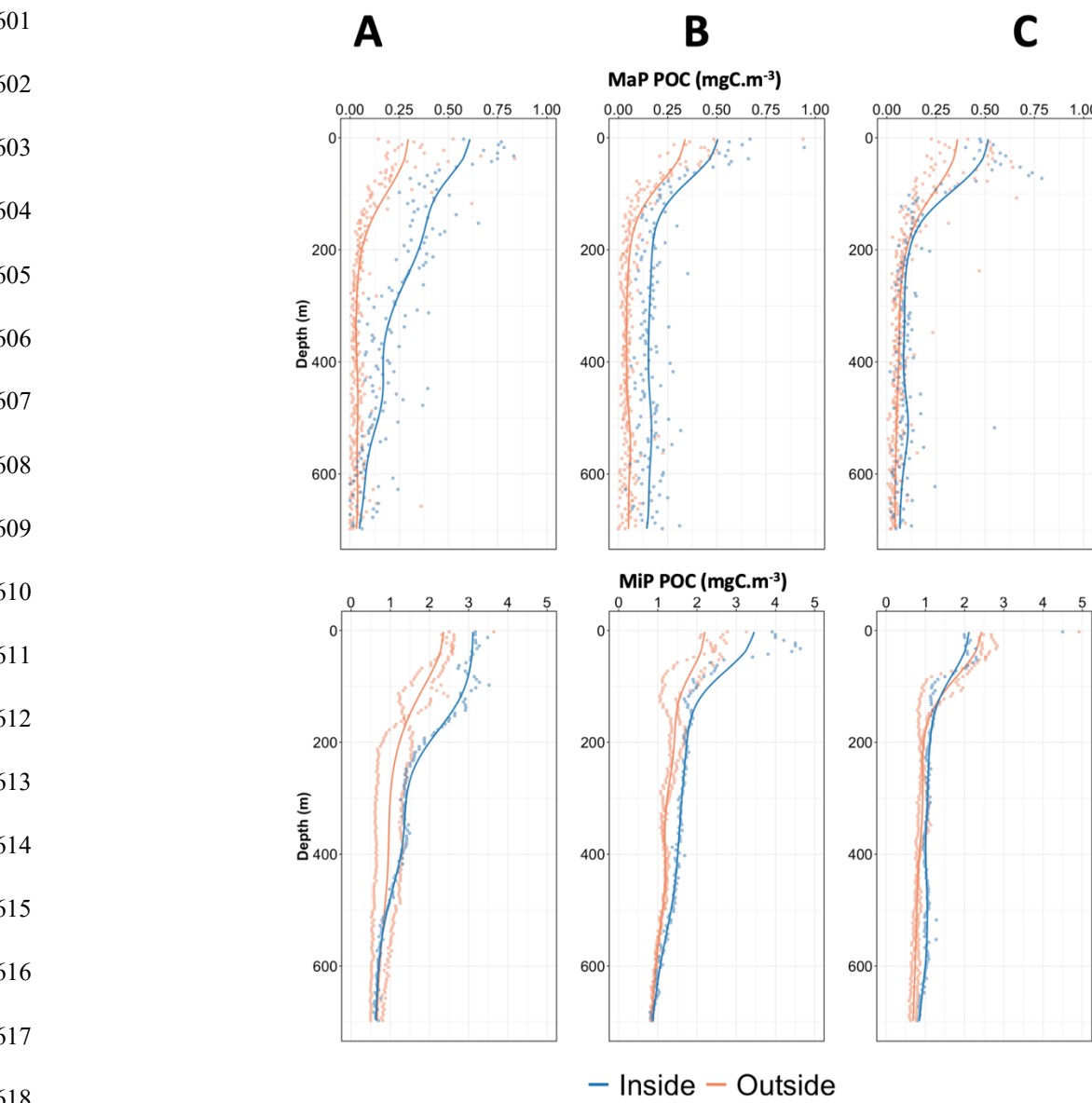

Figure 4: Comparison of MiP POC (top panels) and MaP POC (bottom panels) in the water column both outside and inside each export feature. The panels show, (A) the first export feature (01-10-2021 – 17/10/2021), (B) the second one (01-12-2021 – 19/12/2021) and (C) the last one (01/03/2022 – 28/03/2022). The dots on the graph represent the data averaged over 5-meter bins. The solid lines are a moving average of the 5-meter bins data. The orange signal corresponds to the average of profiles recorded one month prior to and after each feature (outside), while the blue signal corresponds to the average profiles recorded during each feature (inside).

624

625

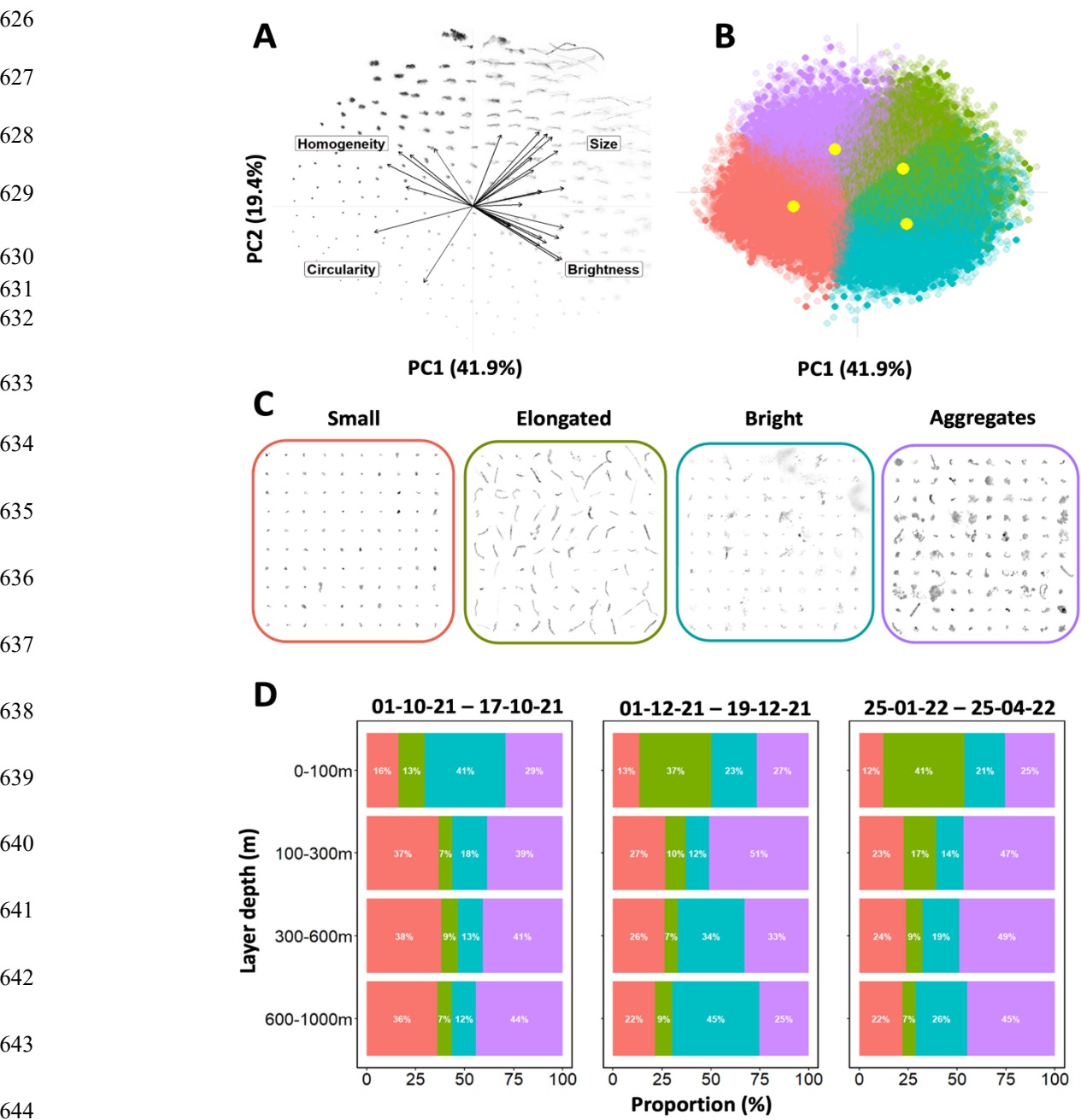

**Figure 5: Results of PCA/K-means clustering on morphological traits of particles. Panels show, (A) the distribution of particle images in the morphospace built by PCA, (B) the K-means clustering classification where each point represents an image, and each cluster is colored independently. The yellow dots represent the center of each cluster. The most transparent dots represent images not retained in the analysis, (C) Representative subset of each morphotype and (D) the proportion (%) of each group according to depth layers during the three export features observed in the morphotypes spatio-temporal distribution. The color code is the same for (B), (C) and (D).**

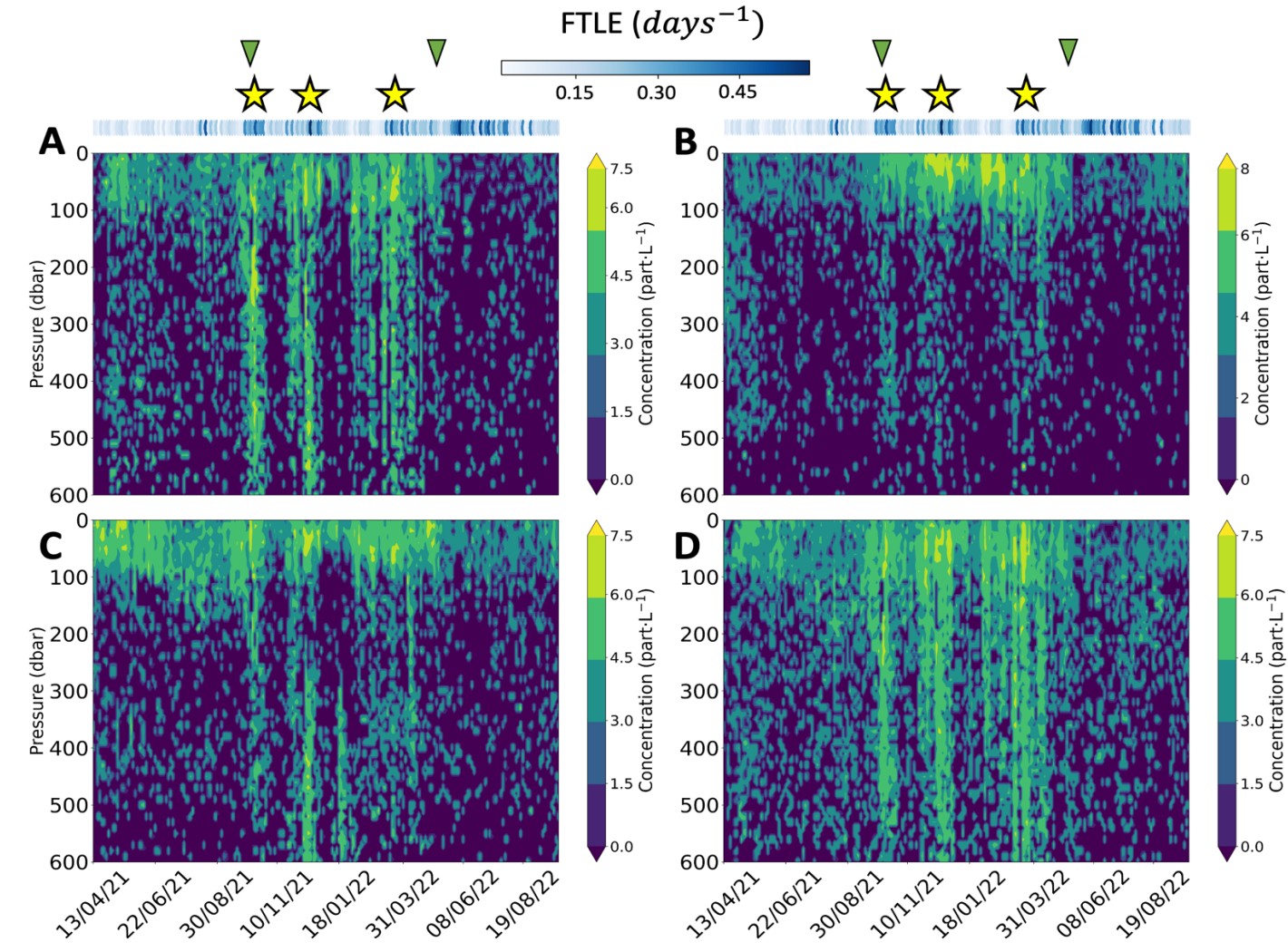

Figure 6: Morphotypes spatio-temporal distribution of exclusive members. The panels show log transformed concentrations (*part.L⁻¹*) of (A) Small, (B) Elongated, (C) Bright and (D) Aggregates morphotypes. Yellow stars and green triangles have the same meaning as in the Figure 2

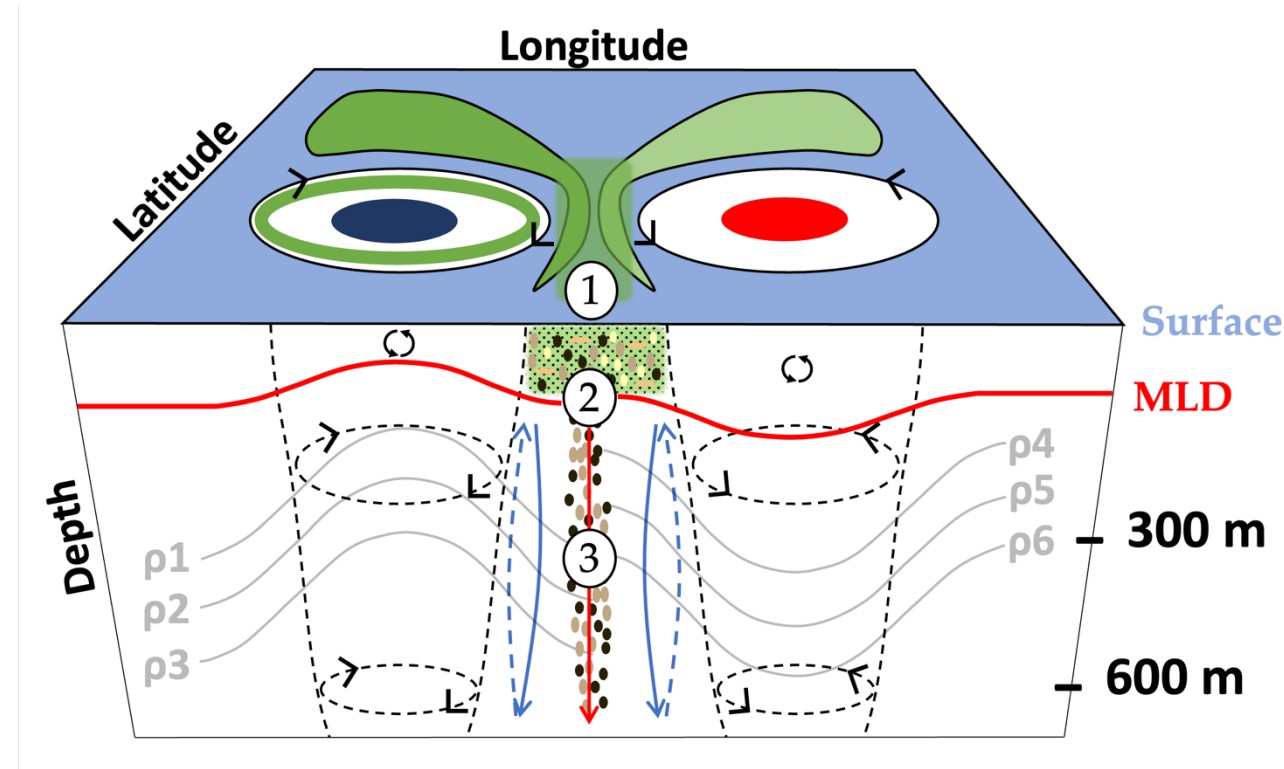

**Figure 7: Schematic view of the proposed mechanism for the deep observed accumulation of particulate carbon in the mesoscale context of the study. (1) Aggregation and coagulation of surface organic matter (green shapes around eddies), facilitated by eddy trapping or stirring. The green circle in the cyclone represents the accumulation of POM at its periphery (resulting from the divergence of the water masses) (2) Influence of the Gravitational Carbon Pump (represented by the red vertical arrow) that transporting selected marine snow types with sufficient downward speed beneath the Mixed Layer Depth (MLD). (3) Coupling with a frontogenesis mechanism inducing enhanced physical vertical speeds (represented by blue arrows), particularly below the MLD and in interface zones between mesoscale structures. The coupling between (1), (2) and (3) can lead to the transport of particles down to significant depths (600 meters in our study). Black arrows in the MLD represent the physical mixing.**

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
