# Peer review of "Intense and localized export of selected marine snow types at eddy"

_EGUsphere, 2024_

## Referee Comment (RC1)

**General comment**

The present study entitled Massive and localized export of selected marine snow types at eddy edges in the South Atlantic Ocean, aims to describe the particles dynamic across physical feature induced by mesoscale eddies in Cape Basin (southwest of Africa). This study is based on BGC-Argo float data acquisition and more specifically the implementation of UVP6 images. The eddy distribution dynamics in the area occurring during the float deployment (17 months) has been characterized by the TOEddies algorithm (The Ocean Eddy Detection and Tracking Algorithm, Laxenaire et al., 2018).

The results presented in this study suggest that particle injection pump is induced by a frontogenesis-driven mechanism (edge of dipole eddy structure) and has the potential to boost the biological pump efficiency.

Globally the manuscript and presented results are clear and easy to get in. The phrasing is straight forward. However, some aspects could have been more carefully checked before the submission such as the figure indexation. The figure numbers in the text don't correspond to the figures presented… One figure (Figure 4) is shown but not presented and described in the text. It's a real pity to submit a manuscript with such errors. I would have expected more details and discussion about the different marine snow aggregates type (shape, density…). Because 'selected marine snow types' is mentioned in the title, I think some input can be done in the manuscript.

I don't require extra analysis in the manuscript. Nevertheless, the quality of the manuscript/figures could be considerably improved. **I recommend moderate revision of the manuscript.**

Please find below my comments and suggestions on the manuscript.

| Principal criteria | Excellent (1) | Good (2) | Fair (3) | Poor (4) |
|---|---|---|---|---|
| **Scientific significance:** Does the manuscript represent a substantial contribution to scientific progress within the scope of Biogeosciences (substantial new concepts, ideas, methods, or data)? | | X | | |
| **Scientific quality:** Are the scientific approach and applied methods valid? Are the results discussed in an appropriate and balanced way (consideration of related work, including appropriate references)? | | X | X | |
| **Presentation quality:** Are the scientific results and conclusions presented in a clear, concise, and well-structured way (number and quality of figures/tables, appropriate use of English language)? | | | X | X (could be really improved) |

Does the paper address relevant scientific questions within the scope of BG? **Yes**

Does the paper present novel concepts, ideas, tools, or data? **No, recent publications about ESP (Eddy Subduction Pump) have already highlighted such as processes on the biological carbon pump. The use of UVP5/6 imaging classification (e.g. elongated aggregates, dark solid aggregates, light aggregates…) is already know as an indicator of particles dynamic phenology.**

Are substantial conclusions reached? **Yes**

Are the scientific methods and assumptions valid and clearly outlined? **Yes**

Are the results sufficient to support the interpretations and conclusions? **Yes, but could be improved (see my comments and suggestions)**

Is the description of experiments and calculations sufficiently complete and precise to allow their reproduction by fellow scientists (traceability of results)? **Yes, if the author provides supplement data (raw data from Argo-Floats and UVP6 images), considering the FAIR protocol.**

Do the authors give proper credit to related work and clearly indicate their own new/original contribution? **Yes**

Does the title clearly reflect the contents of the paper? **Yes, but the use of the term 'Massive' could be nuanced**

Does the abstract provide a concise and complete summary? **Yes**

Is the overall presentation well structured and clear? **Yes**

Is the language fluent and precise? **Yes**

Are mathematical formulae, symbols, abbreviations, and units correctly defined and used? **Yes**

Should any parts of the paper (text, formulae, figures, tables) be clarified, reduced, combined, or eliminated? **Yes, see my comments/suggestions**

Are the number and quality of references appropriate? **Yes, but some recent studies are missing in the manuscript e.g. about stokes influenced sinking aggregates, or stickiness index of particles)**

Is the amount and quality of supplementary material appropriate? **Yes**

**Title**

Line 1: Title, 'Massive' doesn't sound appropriate. Maybe 'Intense and localized export…' or just 'Localized export…'?

**Introduction**

Lines 35-36: 'The solubility pump transports cold, dense, and DIC-rich waters to the deep ocean, as part of the ocean's overturning circulation, primarily in high-latitude regions' add a reference for this statement

Line 44: 'produced by phytoplankton activity', what type of activity? Must be precise or rephrased

Line 46: 'and zooplankton fecal pellets (Turner, 2015).' You might explain the actual process, repackaging by zooplankton induce by grazing and fecal pellet egestion…

Line 47: 'export about 6 Pg of carbon' this statement corresponds to the global carbon export led by gravitational pump alone, I think. If it's what you meant you must precise 'gravitational export flux'. Actual global BCP estimates are about 5-12 PgC yr-1

Line 51: '10 to 100 m d-1 from laboratory experiments (Shanks and Trent, 1980; Azetsu Scott and Johnson, 1992)' you might find a more recent study that established faster sinking rate of laboratory made aggregates and fecal pellets.

Line 54: 'remove 'hopefully'

Line 59: 'can affect all types of particles, from those suspended in the water column to sinking particle', what is the point here? How do you separate the suspended particles in the water column from sinking particles? This sentence must be rephrased

Lines 61-62: 'patchy distribution of animals' what does it mean? Zooplankton grazing activities? What is the expected process here? (aggregation, fragmentation, 'sloppy feeding', consumption?)

Line 66: 'to submesoscale (1-10 km, hours to days) ocean dynamics', submesoscale dynamics could be smaller than a 1km (e.g: fronts and filaments). You must be sure about what spatial scale dynamic is involved in your study and add a reference to support the definition that you are using.

Lines 67-69: This sentence could be implemented. What specific process occurs inside mesoscale eddies that lead to particles aggregation and export? You should find several studies explaining what processes are involved.

Line 75: define 'frontogenesis' and 'mesoscale activity', is related to vertical mixing? Or horizontal mixing? Mesoscale activity could also be interpreted as inshore/offshore exchanges (coastal influence).

Lines 67-74: It could be more convincing if you input some detail of the important process on BCP induced by eddies here. E.g. Eddy subduction pump (ESP).

Line 77: why '600 meters? And why not the mixing layer horizon, or export depth? We could understand the importance of 600m after reading the entire manuscript, but it sounds inappropriate in the introduction.

Line 82: What is the main question behind this work? What is the Hypothesis? It could be a great help to guide the reader.

**Material and Methods**

Line 88: 'SO283 cruise' which research vessel?

Lines 89-90: 'The float remained within the eddy for about 5 months (Baudena et al. 2023, preprint), I would expect more detail of about the eddy's dynamics during the 5 months (maybe show maps of altimetry anomaly with the float location).
* * *
**Fig. 1:** It is not very clear: If the present map corresponds to the surface altimetry anomaly (ADT) by the date of 18-09-2022 and the black line representing the float trajectory from 04-2021 to 09-2022, it means that the float trajectory could not be directly associated to ADT. This map can be split: one map with the float trajectory (with color scale representing the drifting time). Maybe 2 another one with ADT situation at different date

Did you check if the eddies were stable for this period? You mentioned Line 90 that the eddy disappeared from satellite altimetry. I would expect much more detail of the eddy's dynamic during the float deployment period.

What you display in Fig. 7 ('Snapshot of ADT field with the float trajectory (thick black line) during each export feature (A, B and C).') is very interesting, I think you could put these highlights in figure 1, to give an idea of the actual float trajectory regarding the eddies dipole structure evolution (ADT).

Another map with a smaller zoom could be added to this figure (bigger scale area), to have a better idea of the sampling location in the Atlantic Ocean (African East coast).

Finally, what does the red and black (solid and dotted line) represent? I suppose they represent cyclonic/anticyclonic eddies. It should be mentioned in the figure caption.
* * *
Line 126: 'taxonomic identification of macrozooplankton and large particle classification could be conducted.' How? Random forest algorithm? Neural network? Ecotaxa application. Maybe add a short sentence explaining the pre-classification

Line 135: 'The 0.5 mm threshold was used as detrital aggregates > 0.5 mm are considered marine snow (i.e., aggregates, Alldredge and Gotschalk, 1988).' What about the living organisms smaller than 0.5mm (microzooplankton, small copepods)?

Lines 156-157: 'Subsequently, for each group, the first quartile distance was computed, and only individuals with a distance smaller than the first quartile were included in the selection' does it mean that particle images ('exclusive members') out of the first quartile distance were not computed?

Line 158: 'were then used as indicators to see potential differences in the distribution pattern of each morphotype' it is not very clearly explained like this.

Line 182: 'masses—a critical', must be rephrased

**1.4 Lagrangian diagnostics**: This paragraph is very hard to get into.

Lines 207-209: 'A front is defined as a physical barrier that separates two adjacent water masses that have been widely separated in the preceding days and are likely to have different hydrographic properties.' What hydrographic properties? Sea Surface Salinity? Sea Surface

temperature? Sea surface current? Altimetry? FTLEs? It's quite confusing. Such physical barrier must be defined here.

**Results**

Lines 218-219: 'mesoscale activity ', what mesoscale activity, FTLE distribution? Or hydrological variables?
* * *
**Fig. 2**: To improve the readability, the date in x-axis (bottom of C and F) should be bigger. The blue dots (FTLE, top of A and D) could also being bigger.

Would it be possible to add extra text on this figure, describing the relative position regarding the eddies? (e.g. cyclonic eddy, frontal area and anticyclonic eddy)
* * *
Lines 231-232: 'with very different hydrographic characteristics compared with the Benguela upwelling cyclones', what are these very different hydrographic characteristics?

Lines 259-260: 'There was also an observable increase in the concentration of small, labile, and refractory particles in the surface, as reflected by the *bbsr* coefficient (Fig. 2.E)' how can the *bbsr* coefficient could be associated with small, labile, and refractory particles?
* * *
**Fig. 3**: Same comment as for Fig. 2

The **Fig. 4** is described nowhere in the text.

**Fig. 5**: The particles images on the panel C are very small and 'un-contrasted', I know that UVP images are like that (and the contrast is a critical parameter for any classification), but to 'illustrate' the clustering, could you increase the size and the contrast in this panel?

In panel D, 3 columns (stack plot) are displayed (corresponding to the three export features) could you maybe add subtitles with the date of these features?

**Fig. 6**: Same comment as for Fig. 2.

You mentioned the 'White dashed lines' in the caption, because these white dashed lines are not displayed, and would overlap the distribution, please remove it from the caption.
* * *
Line 294: 'except perhaps in the case of column 2.' Avoid the term 'perhaps', the column 2 clearly evidences that 'Bright' particles increase below 300m depth.
* * *
**Fig. 7**: I would expect to see this figure sooner in the manuscript. This figure better explains the float trajectory and record considering the eddies distribution.
* * *
Line 297: 'During the second feature (Fig. 8),'. I think it's still on Fig. 7 (Fig. 7B)

Line 302: 'the southern edge of a large anticyclone (Fig. 9)'. I think it's still on Fig. 7 (Fig. 7C)

Line 308: You mentioned 'POC', how can you be sure it's only POC (and not PIC or other dust deposition). Did you compare the vertical fluctuation of *bbsr* and Chl *a*?

**Discussion**

Line 328: there is no figure 10.1. Do meant Fig. 8?

Lines 331-333: Is it possible that these elongated particles are fecal pellet (stick shaped fecal pellet)?

Lines 334-335: 'Coagulation is responsible for the production of large particles when particle concentrations become important in the water column (Jackson et al.1990)', The coagulation is indeed driven by high particles concentration but mostly by the particle's stickiness and size spectra heterogeneity. You might add an appropriate reference here.

Line 339: Re-check the figures number.

Line 342-343: 'However, typical mesozooplankton fecal pellets (a few hundred μm) were not observed in abundance in any of the four morphological categories.' Is it not the case for elongated aggregates?

Line 343-344: '…zooplankton abundance increases during the three particle distribution events (see Figure S6).' Fig. S6 does not correspond to this statement. Inside the eddy, the copepods abundance seems to be higher (at least for 2 events). Moreover, have you also considered the other zooplankton groups based on the UVP6? (Even If I assume that copepods are the main represented group).

Lines 344-345: 'In our case, it is more likely that physical coagulation had a greater influence on aggregate formation than trophic biological aggregation', please reconsider the higher copepods abundance in the eddy.

Lines 395-396: 'types that we hypothesized to be large phytoplankton', what are approximately the length/width of these elongated particles? What chain forming diatoms groups could be represented?

**Conclusion**

Line 429: Fig.8? (no figure 10)

**Fig. 8**: Why the MLD (red solid line) is equal everywhere (cyclonic eddy, front and anticyclonic eddy? According to the Figure 2, it is evident that the MLD vary between these different physical features

Why the 'particles' (colored dots) are characterized by different colors (4 colors) in the ML but only 2 colors below the MLD? Does it mean something? If yes precise

Explain the green circle is displayed only in the cyclonic eddy

What represent the 2 green shapes around both eddy structures?

---

## Author Comment (AC2)

**Supplementary figures**

[Figure]

Figure S1: Evolution of MaP concentration averaged between 150- and 600-meters depth. Three features were detected using the STARS method. Each dot represents a profile and the blue, red and green ones correspond to the first, second and third feature respectively. The horizontal black line is the MaP concentration mean over the entire deployment. The blue, red and green lines are the mean concentration of each corresponding feature.

[Figure]

[Figure]

**Figure S2: Spearman correlation between averaged MaP concentration and Lagrangian chlorophyll-a (15 days backward) in the upper 100 meters of the water column. The green dots represent the mean of each profile (n = 183). The green line represents the linear regression between both variables. The green shaded area represents the regression confidence interval (95th percentile).**

[Figure]

n = 183
Regression equation: y = 682.11x + -117.86
R-squared: 0.40
p-value: 9.59e-22
Correlation coefficient: 0.65

**Figure S3: Spearman correlation between averaged elongated morphotype concentration and chlorophyll-a concentration in the upper 100 meters of the water column. The green dots represent the mean of each profile (n = 183). The green line represents the linear regression between both variables. The green shaded area represents the regression confidence interval (95th percentile).**

[Figure]

**Figure S4: Spearman correlation between averaged MaP concentration and Lagrangian FTLE (5 days backward) between 100- and 1000-meters depth and during the productive period (from 05/10/2021 to 23/04/2022). The blue dots represent the mean of each profile (n = 66). The blue line represents the linear regression between both variables. The blue shaded area represents the regression confidence interval (95th percentile).**

[Figure]

Figure S5: Spearman correlation between averaged MiP concentration and Lagrangian FTLE (5 days backward) between 100- and 1000-meters depth and during the productive period (from 05/10/2021 to 23/04/2022). The blue dots represent the mean of each profile (n = 66). The blue line represents the linear regression between both variables. The blue shaded area represents the regression confidence interval (95th percentile).

[Figure]

**Figure S6: Spearman correlation between averaged MiP concentration and Apparent Oxygen Utilization (AOU) between 100- and**
**1000-meters depth and during the productive period (from 05/10/2021 to 23/04/2022). The blue dots represent the mean of each**
**profile (n = 66). The blue line represents the linear regression between both variables. The blue shaded area represents the regression**
**confidence interval (95th percentile).**

[Figure]

Figure S7: Comparison of Copepods concentration in the water column both outside and inside each export feature. The panels show, (A) the first export feature (01-10-2021 – 17/10/2021), (B) the second one (01-12-2021 – 19-12-2021) and (C) the last one (01/03/2022 – 28/03/2022). The dots on the graph represent the data averaged over 5-meter bins. The solid lines are a moving average of the 5-meter bins data. The orange signal corresponds to the average of profiles recorded one month prior to and after each feature (outside), while the blue signal corresponds to the average profiles recorded during each feature (inside).

---

## Author Comment (AC4)

[revised manuscript text omitted]

---

## Author Response (AR1)

**Reviewers comments on «Massive and localized export of selected marine snow types at eddy edges in the South Atlantic Ocean » MS No.: egusphere-2024-1558**
**Reviewer's comments in** black – **Responses in** blue –
**Text added to the manuscript in** green -

**Reviewer #1:**

General comments:

The present study entitled Massive and localized export of selected marine snow types at eddy edges in the South Atlantic Ocean, aims to describe the particles dynamic across physical feature induced by mesoscale eddies in Cape Basin (southwest of Africa). This study is based on BGC-Argo float data acquisition and more specifically the implementation of UVP6 images. The eddy distribution dynamics in the area occurring during the float deployment (17 months) has been characterized by the TOEddies algorithm (The Ocean Eddy Detection and Tracking Algorithm, Laxenaire et al., 2018). The results presented in this study suggest that particle injection pump is induced by a frontogenesis-driven mechanism (edge of dipole eddy structure) and has the potential to boost the biological pump efficiency. Globally the manuscript and presented results are clear and easy to get in. The phrasing is straight forward. However, some aspects could have been more carefully checked before the submission such as the figure indexation. The figure numbers in the text don't correspond to the figures presented… One figure (Figure 4) is shown but not presented and described in the text. It's a real pity to submit a manuscript with such errors. I would have expected more details and discussion about the different marine snow aggregates type (shape, density…). Because 'selected marine snow types' is mentioned in the title, I think some input can be done in the manuscript. I don't require extra analysis in the manuscript. Nevertheless, the quality of the manuscript/figures could be considerably improved. I recommend moderate revision of the manuscript.

We would like to thank the reviewer for his numerous comments and for the time he has devoted to it. We have addressed them conscientiously. We correct the figures indexation (which should have been done more carefully before submission). We have also tried to better explain the origin of morphotypes (especially elongated ones) but we must admit that without ancillary data it is not possible to speculate too much. Thanks to all the comments, the quality of the manuscript has been greatly improved. You'll find our responses (in blue) and the corresponding changes in the manuscript (in green) following each comment.

| Principal criteria | Excellent (1) | Good (2) | Fair (3) | Poor (4) |
|---|---|---|---|---|
| **Scientific significance:** Does the manuscript represent a substantial contribution to scientific progress within the scope of Biogeosciences (substantial new concepts, ideas, methods, or data)? | | X | | |
| **Scientific quality:** Are the scientific approach and applied methods valid? Are the results discussed in an appropriate and balanced way (consideration of related work, including appropriate references)? | | X | X | |
| **Presentation quality:** Are the scientific results and conclusions presented in a clear, concise, and well-structured way (number and quality of figures/tables, appropriate use of English language)? | | | X | X (could be really improved) |

Does the paper address relevant scientific questions within the scope of BG? Yes

Does the paper present novel concepts, ideas, tools, or data? No, recent publications about ESP (Eddy Subduction Pump) have already highlighted such as processes on the biological carbon pump. The use of UVP5/6 imaging classification (e.g. elongated aggregates, dark solid aggregates, light aggregates…) is already know as an indicator of particles dynamic phenology.

We agree with the reviewer that the techniques used in this study are not completely new. However, this is the first time that UVP6 data from a recovered BGC-Argo float has been analyzed. Furthermore, we suggest through this study that ideas about ESP are not entirely consistent with our results.

Are substantial conclusions reached? Yes

Are the scientific methods and assumptions valid and clearly outlined? Yes

Are the results sufficient to support the interpretations and conclusions? Yes, but could be improved (see my comments and suggestions)

Is the description of experiments and calculations sufficiently complete and precise to allow their reproduction by fellow scientists (traceability of results)? Yes, if the author provides

supplement data (raw data from Argo-Floats and UVP6 images), considering the FAIR protocol.

We have indicated in the manuscript how to access the images and the counts/size measurements of the objects seen by the UVP6 which were used to carry out this study.

Lines 131-132: "All the images used during this work can be found here: https://ecotaxa.obs-vlfr.fr/prj/8801."

Lines 161-162: "The raw data corresponding to this section can be found on the Ecopart platform (https://ecopart.obs-vlfr.fr, project: uvp6_sn000101lp_2021_WMO6903095_recovery)"

Do the authors give proper credit to related work and clearly indicate their own new/original contribution? Yes

Does the title clearly reflect the contents of the paper? Yes, but the use of the term 'Massive' could be nuanced

We thank the reviewer for this contribution, we agree with this comment and modify the title accordingly.

Line 1: "Intense and localized export of selected marine snow types at eddy edges in the South Atlantic Ocean"

Does the abstract provide a concise and complete summary? Yes

Is the overall presentation well structured and clear? Yes

Is the language fluent and precise? Yes

Are mathematical formulae, symbols, abbreviations, and units correctly defined and used? Yes

Should any parts of the paper (text, formulae, figures, tables) be clarified, reduced, combined, or eliminated? Yes, see my comments/suggestions

Are the number and quality of references appropriate? Yes, but some recent studies are missing in the manuscript e.g. about stokes influenced sinking aggregates, or stickiness index of particles)

Is the amount and quality of supplementary material appropriate? Yes

Comments on the manuscript:

**Title:**
Line 1: Title, 'Massive' doesn't sound appropriate. Maybe 'Intense and localized export…' or just 'Localized export…'?

We change the title accordingly, the new one is:
Line 1: "Intense and localized export of selected marine snow types at eddy edges in the South Atlantic Ocean"

**Introduction:**

Lines 35-36: 'The solubility pump transports cold, dense, and DIC-rich waters to the deep ocean, as part of the ocean's overturning circulation, primarily in high-latitude regions' add a reference for this statement

We added the following reference:
Line 34: "(Sarmiento and Gruber, 2006)"

Line 44: 'produced by phytoplankton activity', what type of activity? Must be precise or rephrased

We removed this part of the sentence and we specified it in the next one
Lines 42-44: "Large particles of POC (>500 μm), also called marine snow, can consist of aggregated phytoplankton cells such as diatoms, different types of aggregates resulting from coagulation processes (Le Moigne, 2019), and zooplankton activity, such as phytoplankton grazing and faecal pellet egestion (Turner, 2015)."

Line 46: 'and zooplankton fecal pellets (Turner, 2015).' You might explain the actual process, repackaging by zooplankton induce by grazing and fecal pellet egestion…

We specified the zooplankton activity in the new text.
Line 44: "zooplankton activity, such as phytoplankton grazing and faecal pellet egestion (Turner, 2015)."

Line 47: 'export about 6 Pg of carbon' this statement corresponds to the global carbon export led by gravitational pump alone, I think. If it's what you meant you must precise 'gravitational export flux'. Actual global BCP estimates are about 5-12 PgC yr-1

Indeed, we wrote about the gravitational flux, we modified references accordingly.
Lines 44-46: "This mechanism is estimated, through empirical and food web models, to export about 3 to 10 Pg of carbon per year below the euphotic zone or 100 meters depth depending on the study (Dunne et al., 2005; Henson et al., 2012; Siegel et al., 2014; Bisson et al., 2020)."

Line 51: '10 to 100 m d-1 from laboratory experiments (Shanks and Trent, 1980; Azetsu Scott and Johnson, 1992)' you might find a more recent study that established faster sinking rate of laboratory made aggregates and fecal pellets.

As reviewer suggests we've added newer references and also one recent synthesis.

Lines 49-53: "Laboratory experiments have estimated that the sinking velocity of marine snow range from a few meters to several hundred meters per day (Shanks and Trent, 1980; Azetsu-Scott and Johnson, 1992; Ploug et al., 2008; Laurenceau-Cornec et al., 2015; Cael et al., 2021) and, in a few cases, from in situ imaging time series (Alldredge and Gotschalk, 1988; Stemmann et al., 2002; Trudnowska et al., 2022; Soviadan et al., 2024)"

Line 54: 'remove 'hopefully'

Done.

Line 59: 'can affect all types of particles, from those suspended in the water column to sinking particle', what is the point here? How do you separate the suspended particles in the water column from sinking particles? This sentence must be rephrased

We have simplified the sentence to be more clear such as :

Line 58: "and can affect all types of particles in the water column"

Lines 61-62: 'patchy distribution of animals' what does it mean? Zooplankton grazing activities? What is the expected process here? (aggregation, fragmentation, 'sloppy feeding', consumption?)

We simplified this section of the text by removing the sentence, as it was not crucial to our study. The conclusion remains that these processes are challenging to sample, which is not further developed in our analysis.

Line 66: 'to submesoscale (1-10 km, hours to days) ocean dynamics', submesoscale dynamics could be smaller than a 1km (e.g: fronts and filaments). You must be sure about what spatial scale dynamic is involved in your study and add a reference to support the definition that you are using.

We corrected the submesoscale definition and added the associated references that support it.
Lines 62-63: "to submesoscale (100 m to 10 km, hours to days) ocean dynamics (Thomas et al., 2008; McWilliams, 2016)."

Lines 67-69: This sentence could be implemented. What specific process occurs inside mesoscale eddies that lead to particles aggregation and export? You should find several studies explaining what processes are involved.

We understand the reviewer's preference against citing reviews. We have therefore developed the following descriptions of selected mechanisms relevant to our study.

Lines 64-72: "In particular, the influence of mesoscale eddies on particle production, aggregation and export (Stemmann et al., 2008; Guidi et al., 2012; McGillicuddy, 2016; Stuckel et al., 2017) is a very active area of research because these structures are ubiquitous in the global ocean and are the largest source of ocean variability (Stammer, 1997; Wunsch, 1999). For example, spatial patterns of particles, differentiated by size, have been shown to be associated with anticyclonic circulation (Gorsky et al., 2002; Waite et al., 2016). Upwelling at eddy's interfaces can enhance phytoplankton productivity and particle production (Shih et al., 2020) while downwelling, contributes to oblique transport of dissolved, and particulate (both sinking and suspended) carbon to depth (Stemmann et al., 2008; Guidi et al., 2012) in a process called the Eddy Subduction Pump (ESP, Omand et al., 2015). Vertical transport has been suggested to be increased by sub-mesoscale vertical velocities associated with fronts (Guidi et al., 2007)."

Line 75: define 'frontogenesis' and 'mesoscale activity', is related to vertical mixing? Or horizontal mixing? Mesoscale activity could also be interpreted as inshore/offshore exchanges (coastal influence).

Indeed, this sentence lacked precision, so we clarified it in the text. This part is now at the end of the introduction.

Lines 83-85: "Based on recurrent observations of marine snow concentration "hot spots" at mesopelagic depth during the productive season, we discuss the role of horizontal and vertical circulation at fronts on marine snow production and export to the deep sea."

Lines 67-74: It could be more convincing if you input some detail of the important process on BCP induced by eddies here. E.g. Eddy subduction pump (ESP).

The Eddy Subduction Pump has now been introduced as follows:

Lines 69-71: "while downwelling, contributes to oblique transport of dissolved, and particulate (both sinking and suspended) carbon to depth (Stemmann et al., 2008; Guidi et al., 2012) in a process called the Eddy Subduction Pump (ESP; Omand et al. 2015)."

Line 77: why '600 meters? And why not the mixing layer horizon, or export depth? We could understand the importance of 600m after reading the entire manuscript, but it sounds inappropriate in the introduction.

We agree that without the context of the study, it's not necessarily easy to understand why we're referring to 600 meters. We have therefore modified this section to refer to the mesopelagic zone (in the introduction and also in the abstract).

Line 19: "down to the mesopelagic zone"

Line 83: "at mesopelagic depth"

Line 82: What is the main question behind this work? What is the Hypothesis? It could be a great help to guide the reader.

The main question of this study and the hypothesis are now described at the end of the introduction.

Lines 83-85: "Based on recurrent observation of marine snow concentration "hot spots" at mesopelagic depth during the productive season, we discuss the role of horizontal and vertical circulation at fronts on marine snow production and export to the deep sea."

**Material and Methods:**
Line 88: 'SO283 cruise' which research vessel?

The float was deployed from the Sonne research vessel. We added this information to the text.
Line 88: "During the SO283 cruise, a BGC Argo float (WMO: 6903095) was deployed, from the RV Sonne, on April 14, 2021"

Lines 89-90: 'The float remained within the eddy for about 5 months (Baudena et al. 2023, preprint), I would expect more detail of about the eddy's dynamics during the 5 months (maybe show maps of altimetry anomaly with the float location).

We understand that readers may want to know more about the "life" of this eddy. However it's not the purpose of our study, but it is actually the point of the cited preprint (Baudena et al., 2023, which can be found here https://www.researchsquare.com/article/rs-3014931/v1). Our study is totally focused on the three export features that were detected by the float after he left this cyclone. However, following the reviewer suggestion, we better specified these details in the text as follow:

Lines 90-92: The float remained within the eddy for about 5 months. During this period, the eddy merged with another cyclone (Baudena et al. 2023), until it disappeared from satellite altimetry maps probably due to subduction (Ioannou et al., 2022). A detailed analysis of this timeframe of the dataset is provided in Baudena et al., (2023).

**Fig. 1:** It is not very clear: If the present map corresponds to the surface altimetry anomaly (ADT) by the date of 18-09-2022 and the black line representing the float trajectory from 04- 2021 to 09-2022, it means that the float trajectory could not be directly associated to ADT. This map can be split: one map with the float trajectory (with color scale representing the drifting time). Maybe 2 another one with ADT situation at different date.

We thank the reviewer for this suggestion. We agree that Figure 1 could be confusing. We have therefore modified it to show the float trajectory with a bathymetric background (fig. 1A). We have also added 3 panels to show the mesoscale context of the three export features (snapshots from TOEddies algorithm, fig. 1.B.C.D) described in our study. We would like to thank the reviewer for this comment, which significantly improved the quality of this figure.

Did you check if the eddies were stable for this period? You mentioned Line 90 that the eddy disappeared from satellite altimetry. I would expect much more detail of the eddy's dynamic during the float deployment period.

Thanks to the TOEddies algorithm, we have access to extensive information about eddies. The three snapshots added show the dynamics of the three export events while maintaining a clear message. In addition, the eddy mentioned above (line XX now) is outside the scope of our study and is the subject of a separate study (Baudena et al. 2023, preprint).

What you display in Fig. 7 ('Snapshot of ADT field with the float trajectory (thick black line) during each export feature (A, B and C).') is very interesting, I think you could put these highlights in figure 1, to give an idea of the actual float trajectory regarding the eddies dipole structure evolution (ADT).
Done

Another map with a smaller zoom could be added to this figure (bigger scale area), to have a better idea of the sampling location in the Atlantic Ocean (African East coast).
Done

Finally, what does the red and black (solid and dotted line) represent? I suppose they represent cyclonic/anticyclonic eddies. It should be mentioned in the figure caption.
Done

Line 126: 'taxonomic identification of macrozooplankton and large particle classification could be conducted.' How? Random forest algorithm? Neural network? Ecotaxa application. Maybe add a short sentence explaining the pre-classification

We clarified the text by adding:
Lines 127-131: "taxonomic identification of macrozooplankton and large particle classification could be conducted on the Ecotaxa platform (http://ecotaxa.obs-vlfr.fr). Taxonomic classification was initially assisted by a CNN (Convolutional Neural Network) algorithm to extract descriptive features from the images. From these, the taxonomic group of each image was predicted using a Random Forest algorithm. Then, living organisms' predictions have all been manually validated by taxonomic experts."

Line 135: 'The 0.5 mm threshold was used as detrital aggregates > 0.5 mm are considered marine snow (i.e., aggregates, Alldredge and Gotschalk, 1988).' What about the living organisms smaller than 0.5mm (microzooplankton, small copepods)?

We thank the reviewer for raising this important point. The separation between MiPs and MaPs was made for visualization purposes, and the 0.5 mm threshold was chosen as marine snow is normally considered greater than that. However, the counts/size measurements of the objects seen by the UVP6 (from the Ecopart platform) contain zooplanktons and images (for objects ESD ≥ 500µm) are saved and processed in an independent way (on the Ecotaxa platform). Furthermore, it is not possible to determine whether an object smaller than 0.5 mm is a living organism or detritus due to the limited resolution of the image (they are not even saved during the deployment). As a result, the MiP and MaP signals could be influenced by zooplankton abundances. However, the proportion of living organisms observed by the UVP6 compared to detritus is low (approximately 20% in our study for particles >500µm). The zooplankton contribution of smaller objects may even get smaller as size decreases (Stemmann and Boss, 2012). This could introduce a bias in MiP counts which could contain zooplankton organisms. We have addressed and specified in the text.

Lines 141-146: "The 0.5 mm threshold was used as detrital aggregates > 0.5 mm are considered marine snow (i.e., aggregates, Alldredge and Gotschalk, 1988) and was chosen for visualization purposes. However, the count/size measurements of the objects seen by the UVP6 contain zooplanktons and it is not possible, from the images, to determine whether an object smaller than 0.5 mm is a living organism or detritus due to the limited resolution. As a result, the MiP and MaP signals could be influenced by zooplankton abundances. However, the proportion of living organisms to the total particle counts (ESD > 500µm) are usually smaller than 10% (Stemmann and Boss, 2012).

Lines 156-157: 'Subsequently, for each group, the first quartile distance was computed, and only individuals with a distance smaller than the first quartile were included in the selection' does it mean that particle images ('exclusive members') out of the first quartile distance were not computed?

See below the answer.

Line 158: 'were then used as indicators to see potential differences in the distribution pattern of each morphotype' it is not very clearly explained like this.

Here we answer the two previous comments.

Lines 182-185: "Subsequently, for each group, the first quartile of the Euclidean distance distribution was computed, and only individuals with a distance smaller than the first quartile were included in the selection. The concentrations of the different morphotypes shown in figure 6 correspond to those of these "exclusive members". Objects with a too large distance (out of the first quartile) were not included in the further analysis."

Line 182: 'masses—a critical', must be rephrased

We rephrased as follow:
Lines 216-217: "the origin of sampled water masses which is a critical component for the objectives of this study"

**1.4 Lagrangian diagnostics:** This paragraph is very hard to get into.

Lines 207-209: 'A front is defined as a physical barrier that separates two adjacent water masses that have been widely separated in the preceding days and are likely to have different hydrographic properties.' What hydrographic properties? Sea Surface Salinity? Sea Surface temperature? Sea surface current? Altimetry? FTLEs? It's quite confusing. Such physical barrier must be defined here.

Following the remark by Reviewer 1, we have modified the paragraph of presentation of the FTLE calculation and properties. We have tried to clarify in the most simple and straightforward way what Lyapunov Exponents identify. Basically, they are computed in such a way that intense FTLE values identify filament-like structures that separate water masses that were very distant in the previous days (and that are now adjacent). As these water masses come from distant locations, they are very likely to be very different in terms of properties. Indeed, previous studies were able to show that intense FTLE ridges separate water masses that often are contrasted in terms of temperature, chlorophyll, and even community composition. Following the hint from Reviewer 2, we also added a sentence about the fact that FTLE ridges (*fronts*), as they represent a zone of convergence between two water masses, are also considered as proxies of regions with intense vertical velocities. The paragraph now looks as it follows:

Lines 241-247: "The first diagnostic we implemented was the Finite-Time Lyapunov Exponent (FTLE, days-1; Shadden et al., 2005). By construction, intense FTLE values (typically disposed in filament shapes) are found at the edge between two water masses that have been widely separated in the preceding days. As these two water masses come from distant locations, they are likely to share different hydrological characteristics, such as temperature, primary production or biological activity (d'Ovidio et al., 2010; Haller, 2015; Lehahn et al., 2018). For these reasons, FTLE is useful to identify oceanic fronts and can be considered as proxies for water masses convergence and thus, possibly, associated vertical downwelling (McWilliams, 2016)."

**Results**
Lines 218-219: 'mesoscale activity ', what mesoscale activity, FTLE distribution? Or hydrological variables?

Here we were referring to eddies activity, so we specified it in the text

Lines 255-256: "The lateral variability in thermohaline properties (Fig. 2) is driven by mesoscale eddies and submesoscale features, which carry distinct signatures and interact dynamically, creating spatio-temporal heterogeneity in water masses."

**Fig. 2:** To improve the readability, the date in x-axis (bottom of C and F) should be bigger. The blue dots (FTLE, top of A and D) could also being bigger.
Done

Would it be possible to add extra text on this figure, describing the relative position regarding the eddies? (e.g. cyclonic eddy, frontal area and anticyclonic eddy)

We thank the reviewer for his suggestion regarding adding information about the relative position of eddies to the figure. While we understand the importance of highlighting these features, we feel that the current figure is already visually dense. Adding additional text could compromise both the readability and the overall clarity of the figure.

Lines 231-232: 'with very different hydrographic characteristics compared with the Benguela upwelling cyclones', what are these very different hydrographic characteristics?

Indeed this sentence lacked precision. Giulivi and Gordon (2006) observed an intrusion of warmer, saltier Indian Ocean water in the south, driven by the spawning of Agulhas Rings at the Agulhas Current retroflection. Since Indian Ocean water is characteristically warmer and saltier, the water within cyclonic eddies originating from these regions exhibits relatively warmer and saltier anomalies compared to similar features in the northerly region of the South-East

Atlantic Ocean. We have not gone into detail in the manuscript, as this is not the central focus of our study. However, we have clarified the difference in properties in the text.

Lines 267-270: "This type of cyclone originates from the southern African continental slope from barotropic instabilities and enters the ocean interior at the northern edge of the Agulhas Current Retroflection (Duncombe Rae, 1991) with very different hydrographic characteristics (warmer and saltier water masses, Giulivi and Gordon, 2006) compared with the Benguela upwelling cyclones."

Lines 259-260: 'There was also an observable increase in the concentration of small, labile, and refractory particles in the surface, as reflected by the bbsr coefficient (Fig. 2.E)' how can the bbsr coefficient could be associated with small, labile, and refractory particles?

As described in section 1.1.1, the backscattering signal was decomposed to extract the signal corresponding to the smallest particles. This method is the one proposed by Briggs et al., 2020. We clarified it in the text. We changed "small labile and refractory particles" to "small particles" to avoid confusion and added info on the Briggs et al. 2020 method.

Lines 118-122: "The backscattering coefficient signal (bbp) was decomposed to extract the signal of small particles (bbsr) from the raw signal which also contained spikes triggered by rare large aggregates passing in the flow field. This was done by applying the method proposed by Briggs et al., 2020 (see supplementary materials) that decomposes the backscattering signal into its baseline (as a proxy of small particles, *bbsr*) and intermittent pulses (as proxy of larger aggregates)"
* * *
**Fig. 3:** Same comment as for Fig. 2

Done

The **Fig. 4** is described nowhere in the text.

Figure 4 is already described in the 2.6 section.
Lines 342-345: "Figure 4 clearly illustrates the high abundance of particulate carbon associated with the three export events, with their influence extending down to 600 m depth. The abundance increased by a factor of 2-3 (with a factor of more than 7 during the first feature) compared to periods when the Argo float was moving 'outside' these frontal regions. An average increase in MiP POC of about 25% is also observed for the first and the second features"

**Fig. 5:** The particles images on the panel C are very small and 'un-contrasted', I know that UVP images are like that (and the contrast is a critical parameter for any classification), but to 'illustrate' the clustering, could you increase the size and the contrast in this panel?
Done
In panel D, 3 columns (stack plot) are displayed (corresponding to the three export features)
could you maybe add subtitles with the date of these features?
Done
**Fig. 6:** Same comment as for Fig. 2.
Done
You mentioned the 'White dashed lines' in the caption, because these white dashed lines are
not displayed, and would overlap the distribution, please remove it from the caption.

Done.

Overall, we thank the reviewer for these suggestions, which have greatly contributed to improving the quality of the figures.

Line 294: 'except perhaps in the case of column 2.' Avoid the term 'perhaps', the column 2 clearly evidences that 'Bright' particles increase below 300m depth.

We removed "perhaps".

**Fig. 7:** I would expect to see this figure sooner in the manuscript. This figure better explains the float trajectory and record considering the eddies distribution.

Figure 7 is now merged with figure 1

Line 297: 'During the second feature (Fig. 8),'. I think it's still on Fig. 7 (Fig. 7B)

The reviewer is right, we corrected the figure indexation.

Line 302: 'the southern edge of a large anticyclone (Fig. 9)'. I think it's still on Fig. 7 (Fig. 7C)

The reviewer is right again, we corrected the figure indexation.

Line 308: You mentioned 'POC', how can you be sure it's only POC (and not PIC or other dust deposition). Did you compare the vertical fluctuation of bbsr and Chl a?

We thank the reviewer for this comment. It is unlikely that these are dust particles, as the study area is not prone to significant dust deposition. While it is possible that particulate inorganic carbon (PIC) may be present, for example, if a pelagic *Foraminifera* is incorporated into a marine snow particle. However, to address this potential issue we used a formula dedicated to estimates POC content for miscellaneous marine snow, including components such as fecal pellets. We added text in the Materials and Methods section to explain how carbon contents were estimated.

Lines 147-153: "Organic carbon content of MiPs and MaPs was calculated assuming that particle mass can be estimated using an empirically derived relationship for marine aggregates (Alldredge, 1998; Kriest, 2002 reference 2a of Table 1). Assuming a carbon:nitrogen ratio of 106:16, this yields an expression for the carbon content of a particle in a given size class characterized by its diameter ESD (in cm). Moreover, this formula is dedicated to estimate POC content of miscellaneous marine snow, including components such as fecal pellets. Multiplying the carbon content with the particle number in this size class (in particles.m$^{-3}$), and integrating over the MiP and MaP size classes, respectively, we obtain the total POC content (mgC.m$^{-3}$) for MiPs and MaPs."

**Discussion**
Line 328: there is no figure 10.1. Do meant Fig. 8?

The reviewer is right again, we corrected the figure indexation.

Lines 331-333: Is it possible that these elongated particles are fecal pellet (stick shaped fecal pellet)?

We clarified this point with the following text.

Lines 369-376: "Their origin is difficult to determine. Their filamentous morphology suggests a resemblance to chain-forming diatom colonies. The average size of particles classified under the 'elongated' morphotype is approximately 820 μm (±96 μm). Given their size and the study area, it's likely that some of these particles include colonies of *Chaetoceros sp.*, which are known to form filamentous structures and to dominate phytoplanktonic blooms during the austral summer (Laubscher et al., 1993). They could also be *Euphausiid* fecal pellets, however, this kind of organisms was not observed at the surface by the UVP6 which could be attributed to their potential avoidance behavior, making their identification challenging. However, given their size (>500 μm), it is unlikely that they are copepod fecal pellets, most of which are smaller than 200 μm (Møller et al., 2011)."

Lines 334-335: 'Coagulation is responsible for the production of large particles when particle concentrations become important in the water column (Jackson et al.1990)', The coagulation is indeed driven by high particles concentration but mostly by the particle's stickiness and size spectra heterogeneity. You might add an appropriate reference here.

We've added appropriate references as suggested by the reviewer.

Lines 377-380: "Coagulation is responsible for the production of large particles when both particle concentrations and stickiness are high. Additionally, a heterogeneous particle size distribution enhances coagulation, as smaller particles have a higher probability of colliding with larger particles (Hunt, 1982; McCave, 1984; Jackson, 1990)."

Line 339: Re-check the figures number.
Done

Line 342-343: 'However, typical mesozooplankton fecal pellets (a few hundred μm) were not observed in abundance in any of the four morphological categories.' Is it not the case for elongated aggregates?

We have removed this sentence from the section as it was repetitive. The reviewer may refer to our previous response on this topic for further details.

Line 343-344: '…zooplankton abundance increases during the three particle distribution events (see Figure S6).' Fig. S6 does not correspond to this statement. Inside the eddy, the copepods abundance seems to be higher (at least for 2 events). Moreover, have you also considered the other zooplankton groups based on the UVP6? (Even If I assume that copepods are the main represented group).

We have clarified the text and mentioned the cases when copepods were high. However, this was not significant.

Lines 388-390: "We haven't also observed a clear increase in zooplankton abundance during the three particle distribution events. However, a slight increase in copepod abundance was noted during the first and third features within the first 100 meters (see Figure S7)."

Lines 344-345: 'In our case, it is more likely that physical coagulation had a greater influence on aggregate formation than trophic biological aggregation', please reconsider the higher copepods abundance in the eddy.

Lines 390-391: "In our case, it is more likely that physical coagulation had a greater influence on aggregate formation, however, we cannot rule out trophic biological aggregation"

Lines 395-396: 'types that we hypothesized to be large phytoplankton', what are approximately the length/width of these elongated particles? What chain forming diatoms groups could be represented?

The average size of particles belonging to the 'elongated' morphotype is about 820 µm (+/- 96 µm). Given the study area and their size, it is possible that this morphotype is partly composed of diatom colonies (*Chaetoceros sp.*), which are known to be filamentous and to dominate phytoplankton blooms during austral summer. We specified this hypothesis in the text.

Lines 379-373: "Their origin is difficult to determine. Their filamentous morphology suggests a resemblance to chain-forming diatom colonies. The average size of particles classified under the 'elongated' morphotype is approximately 820 µm (±96 µm). Given their size and the study area, it's likely that some of these particles could include colonies of *Chaetoceros sp.*, which are known to form filamentous structures and to dominate phytoplanktonic blooms during the austral summer (Laubscher et al., 1993).

**Conclusion**
Line 429: Fig.8? (no figure 10)
This figure is now Figure 7.
* * *
**Fig. 8:** Why the MLD (red solid line) is equal everywhere (cyclonic eddy, front and anticyclonic eddy? According to the Figure 2, it is evident that the MLD vary between these different physical features

Having a constant depth for the MLD was for simplicity and we have now modified its depth to be qualitatively deeper in the anticyclone and shallower in the cyclone (figure 7 in the reviewed manuscript).

Why the 'particles' (colored dots) are characterized by different colors (4 colors) in the ML but only 2 colors below the MLD? Does it mean something? If yes precise

We specified in the caption that only certain types of marine snow, with sufficient sinking speed, can cross the MLD. This is why only two colors are shown below the MLD (line 714).

Explain the green circle is displayed only in the cyclonic eddy

The green circle in the cyclone represents the accumulation of POM at its periphery (resulting from the divergence of the water masses). This aspect is now specified in the caption (lines 712-713).

What represent the 2 green shapes around both eddy structures?

They represent patches of surface organic matter. We specified it in the caption (line 711).
* * *
Reviewer #2:

General comments:

The authors provided an interesting and serious observational (BGC-Argo float) evidence of 3 major events of POC (marine snow) injection at great (~600m) depth. The article contributes nicely to the timely topic where physical (sub)mesoscale features can interact with the BCP. The descriptive study is serious and offer a qualitative /observational description.

I really enjoyed the straightforward reading. The overall good structured really helps the reader. In terms of science, the analysis of the POC dynamics is well established. However, the physical oceanography assessment is extremely weak and speculative. I understand that it is impossible to reach out to vertical velocities (either from floats or even models are bad at it), but the only physical diagnostic provided (besides positioning with eddies) is the FTLE, a front detection diagnostic. This is an indicator for water mass convergence (explained too late in the text). Everything else is purely speculative (relying on the extensive literature of frontogenesis of course) and/or descriptive. So I felt quite frustrated as it could have gone a bit further, especially with the Lagrangian diagnostics. I would have been more convincing if, for example, the authors provided more straightforward evidence of upwelling / downwelling. Just drawing some isopycnals on top of the MLD could intuitively help (Fig. 2, 3). More importantly, the authors have to provide (or at least try to) more convincing evidences of the physical subduction. The demonstration is now purely qualitative although there are some reachable techniques to objectively characterize subduction events from BGC argo floats (see for example: https://agupubs.onlinelibrary.wiley.com/doi/full/10.1002/2017JC012861, or https://bg.copernicus.org/articles/18/5539/2021/ ).

We thank the reviewer for the general comments and for providing the two references to characterize objectively possible subduction. One of these references (Llort et al., 2018) was already cited in the manuscript in a qualitative discussion. In the revised version, we applied the algorithm to our data set and found that possible subduction events do not always match our columns of marine snow. We discuss more on the differences in the revised version and a new figure 1 has been modified to display the result of this analysis (panel A). Overall, we think that adding this new analysis has greatly improved the quality of the manuscript and we thank the reviewer for it.

Lines 199-208:
"1.2.6 Eddy subduction detection
To compare our findings with the literature, we applied the algorithm proposed by Llort et al., 2018 to detect ESP events from AOU and spiciness anomalies derived from BGC-Argo float data. Spiciness was defined as in Flament, 2002 and allows differentiating water masses with distinct thermohaline properties but with similar density. Then, we compared each profile's 3-bin smoothed values to a 20-bin average to detect anomalies for both AOU and spiciness. An event was classified as subduction-driven only if AOU and spiciness anomalies occurred at the same depth, which helped differentiate subduction effects from horizontal mixing. Thresholds of AOU anomaly < -8 µmol/kg and spiciness < -0.05 were set to detect these kinds of events. Finally, we compared the ESP occurrence with the export features spatial distribution detected with the STARS method. We also considered calculating vertical velocities according to Siegelman et al. (2020), but the temporal resolution of our data was not sufficient to enable this (pers. communication L. Siegelman)."

Lines 415-428: "Previous studies in a similar mesoscale physical context have shown that mesoscale activity can result in the intrusion of small POM-rich water masses into the mesopelagic layers (Omand et al., 2015; Llort et al., 2018). In our study, the three features described, occurred from the MLD to a depth of about 600 m or deeper (i.e. over a large part of the water column), whereas the subduction described in Llort et al. (2018) occurred well below the MLD in the mesopelagic in depth layers thickness of about 50-100 m. We applied the Llort et al. (2018) algorithm to identify ESP events across all profiles. Only 28 profiles were classified as ESP events (see Figure 1.A, yellow dots). Of these, seven were located near export features described in this study. However, increases in bbsr, chlorophyll-a, MiP, and MaP were not consistently observed across these ESP events. Furthermore, the ESP events were generally less than 100 meters thick, whereas the export features we identified extended through much of the water column. These findings suggest that while some export features may be linked to ESP events, the ESP alone does not appear to be a fundamental trigger for these features. This suggests that we are describing a slightly different mechanism compared to Omand et al. (2015) and Llort et al. (2018) responsible for the accumulation of organic matter at depth, but also one related to mesoscale and submesoscale processes. This highlights the fact that the influence of these physical mechanisms is a challenging area of research."

Lagrangian Chlorophyll a is introduced, mentioned briefly in section 3.1 but never shown (unless I missed the obvious) ! Overall, the article would benefit to better exploit the strength of Lagrangian diags. The authors should do better at explaining with those tools how they can properly rule out this is not lateral advection.

Indeed, we forgot to provide a figure for the Lagrangian chlorophyll-a. We add the supplementary figure S2 that shows the relationship between Lagrangian chlorophyll-a and MaP in the surface and we have modified the text as follows.

Lines 358-359: "Regarding the spatiotemporal distribution of MiP and MaP (Fig. 3.B), the abundance of MaP (i.e. aggregates) in the surface layer (0-100m) was significantly correlated with the surface Lagrangian chlorophyll-a 15 days backward (see Fig. S2)"

Regarding the fact that the columns of particles we observe are simply due to lateral advection, we cannot entirely rule it out due to the nature of our data. Specifically, the Lagrangian diagnostics are calculated using geostrophic currents, thus they refer mainly to the mixed layer. In addition, we do not have 3D information on the water body prior to the passage of the drifter, both in terms of currents and biogeochemical properties, making it impossible to determine what occurred before the float passed through. Nevertheless, the strong correlation between the described events, the FTLE, and the positions of the eddies suggests a significant link to vertical rather than lateral advection. In addition, previous studies have shown that frontal structures identified by Lyapunov Exponents ridges at the surface can persist in the water column (Bettencourt et al., 2012), corroborating this hypothesis. We thank the reviewer for this comment. We have specified this limit in the text.

Lines 475-481: "However, we lack information on the 3D context of the water mass, both in terms of currents and biogeochemical properties, which makes it impossible to determine what happened at depth before the float passed. Consequently, it's impossible to rule out the action of lateral advection on the mechanism described in this study.

Nevertheless, the strong correlation between the described events, the FTLE, and the positions of the eddies suggests a significant link to vertical rather than lateral advection. In addition, previous studies have shown that frontal structures identified by Lyapunov Exponents ridges at the surface can persist in the water column (Bettencourt et al., 2012), corroborating our findings."

Another aspect which was disappointing, is the absence of quantitative assessment. The authors do have the carbon content and sinking rate (correct me I may be wrong), so what does stop them from providing fluxes ? If there is a good reason, then the authors should explain the reviewers.

We thank the reviewer for raising this interesting point. Indeed, this study lacks a quantitative assessment of the vertical POC flux. The literature proposes various methods for estimating it but these approaches often overlook the influence of physical processes on such estimates (they are based solely on a size-sinking speed and carbon content relationships). Our findings suggest the presence of vertical velocities within the water mass, which likely influence the distribution of marine snow. Thus, a simple estimate based on size would not have been representative of the real fluxes of the study zone. Moreover, we are unable to disentangle the effects of particle sinking velocities from the vertical displacement of the water mass. For this reason, we have not provided flux estimates in this study.

Lines 154-160: "Our study suggests the presence of vertical velocities within the water column which could significantly influence the marine snow distribution (MiPs and MaPs) at depth. The literature provides various methods for estimating POC fluxes from imaging devices. However, these approaches are based on particle size, sinking speed, and carbon content relationships which do not include the influence of physical processes such as water masses vertical movement. Therefore, a simple POC fluxes estimates from particle size would not accurately represent the actual fluxes in our study area. Furthermore, we couldn't disentangle the contributions of particle sinking speeds from the vertical displacement of the water mass. Given these limitations, we chose not to provide flux estimates."

Otherwise, it feels that the authors are a bit over-influenced by their own community, lacking a bit of prospective from the global community, as the same references are used over and over. On many occasions, they provided references of reviews, not the actual research, which should be (I hope you agree) avoided. But I guess this is the case of every working groups.

We agree on the comment, but we have tried to limit the number of citations because the manuscript contains more than 90 references which is a lot. This point was also raised by Reviewer 1, particularly in the introduction (specifically regarding the section on eddies-related processes; lines 64 to 74). To address this, we have compiled a more comprehensive bibliography to resolve this problem.

Besides this, the authors should re-work the streamlining of the abstract and introduction, but it is not a major issue. I also noted many little awkward formulations that also should be easily fixed.

With both reviews which are complementary, we have revised substantially the abstract and the introduction. We hope that "awkward formulations" have been removed.

Comments on the manuscript:

Figure 1: It would help to highlights where the export events happened here. Although this is shown in detail in Figure 7.

Done

Figure 7: "Solid lines correspond to the eddy maximum speed" Isnt this just isoline of ADT ? Please specify/correct.

We specify the figure caption (which is the figure 1 now). It's true that the solid lines represent ADT isolines but we also distinguish between thick and thin lines. The thick lines represent the eddy maximum speed.

Lines 547-548: "Thick solid lines correspond to eddies maximum speed. Thin solid lines correspond to ADT isolines"

Line 13: If you are referring to climate change then it should be followed by "uptaking and sequestering anthropogenic CO2 from the atmosphere"

See answer below.

Line 14-16: The sentence reads weirs. I would say that the storage relies on ocean pump. Period. Then the pumps consist of the PCP and BCP…

See answer below.

Line 17: BCP -> Organic carbon right ?

The beginning of the abstract has been changed to be more direct. The sentences quoted in the three previous comments have been removed from the abstract.

I personally find that the first lines 13-18 are a very verbose and a long way to present the article. It really sounds like an introduction. A lengthy definition should not stand in an abstract. I suggest re-working the abstract. This article is more like a process study. The physical pump is off-topic, as well as the climate change aspect. Go straight to your point. What's novel about the article is the physical-biological interaction. Emphasize this instead.

Just an example:

" The transfer of organic carbon from the upper to the deep ocean forms the long-term storage of carbon of biological origin. This 'biological carbon pump' is a critical component of the global carbon cycle, reducing atmospheric CO2 levels by ~200 ppm relative to a world without export flux. "

…

"However, the BCP interactions with physical features like eddies have been largely overlooked. This study highlights…" or something like that

We agree with the reviewer that these lines sound like an introduction. We modified the beginning of the abstract to be more straightforward. We thank the reviewer for his suggestions.

Lines 14-16: "The Biological Carbon Pump (BCP) comprises a wide variety of processes involved in transferring organic carbon from the surface to the deep ocean. This results in long-term carbon sequestration. Without BCP, atmospheric CO2 concentrations would be around 200 ppm higher"

Line 18: Certainly not !! What you are investigating is the export "potential" or something like that. You are not quantifying the export flux, neither the export efficiency (e-ratio), or even less the BCP efficiency.

Read for example: 10.1029/2021GB007083 or 10.1111/gcb.17124

We recognise that this is a misuse of language on our part. We removed this sentence from the abstract and we've added this new one instead.

Lines 16-17: "This study reveals that ocean dynamics at mesoscale and submesoscale could have a major impact on particulate organic matter (POM) vertical distribution."

Line 23: "Lagrangian diagnostics": would delete "physics"

Done.

Line 37: "and operates worldwide" weird formulation… I would remove.

Done.

Line 40: Wrong citation. Overall biased and very influenced literature. I recommend to zoom out a bit to the outside of your community. Boyd et al. certainly did not come up with this number. This number is from the bible of biogeochemistry: Sarmiento, J. L. & Gruber, N. Ocean Biogeochemical Dynamics Ch. 8 (Princeton Univ. Press, Princeton, 2006).

We replaced with the correct quote

Line 45: I would rephrase this more straightforwardly: "Large particles of POC (>500µm) are called marine snow,…"

We rephrased the sentence.

Lines 42-44: "Large particles of POC (>500 µm), also called marine snow, can consist of aggregated phytoplankton cells such as diatoms, different types of aggregates resulting from coagulation processes (Le Moigne, 2019), and zooplankton faecal pellets (Turner, 2015)."

Line 47: quite painful to see all this extensive, creative and quantitative science attributed to a single review paper.. please use original studies on top.

We modified the references accordingly.
Lines 44-46: "This mechanism is estimated, through empirical and food web models, to export about 3 to 10 Pg of carbon per year below the euphotic zone or 100 meters depth depending on the study (Dunne et al., 2005; Henson et al., 2012; Siegel et al., 2014; Bisson et al., 2020)."

Rate = speed ?

Yes, we replaced rate by speed as you proposed.

Lines 53-55: un-worthy repetition. Just say in situ imagery have been proved a useful tool to measure sinking speeds. Period.

We change the text accordingly.

Lines 53-54: "In situ imagery has proven to be a valuable tool for estimating sinking speeds"

Line 57: Organic carbon. Migration pump out of topic.

Although, migration pump is not the topic, we kept the sentence because this section describes very briefly the different processes involved in particle transfer to the deep ocean.

Line 75: I would rather use "interact" rather than "coupled"

This sentence is no longer in the text.

Line 125: "various biogenic matter such as POC (Particulate Organic124 Carbon), marine snow and fecal pellets" weird phrasing… fecal pellets and marine snow is not POC ? To me all this is organic particulate detritus. Therefore it's all POC. I would just say "organic/biogenic detritus" or just "POC". This is confusing.

We agree with your comment, we changed the sentence accordingly.

Lines 125-127: "The UVP6 detects and measures the size (from 0.102 to 16.40 mm in Equivalent Spherical Diameter, ESD) of zooplankton and various organic/biogenic matter such as marine snow and fecal pellets."

Line 137. This is just a comment. Nothing to answer here. I acknowledge that this instrument, the UVP, is a valuable imagery system. But at this extreme level of post-processing, and classification, one could however wonder if the data is still data. But it seems to work !

Line 218: convoluted ! Please use something more straightforward like: "Due to intense mesoscale activity, water masses exhibit intense spatio-temporal variability in the studied area"

We thank the reviewer for this suggestion and reformulated the sentence in order to be more straightforward.

Lines 255-256: "The lateral variability in thermohaline properties (Fig. 2) is driven by mesoscale eddies and submesoscale features, which carry distinct signatures and interact dynamically, creating spatio-temporal heterogeneity in water masses."

Line 258: "evidenced by"

Done.

Line 259-260: try using a more direct formulation. E.g.: "Also, a noticeable increase in bbsr indicates greater concentrations of small… "

Lines 297-298: "Also, a noticeable increase in the bbsr coefficient indicates greater concentrations of small particles in the surface (Fig. 2.E)"

Line 290: No need for "indeed" here.

We removed "indeed".

Line 298: Rvmax is never defined… at least I did not find it.

Sorry there was a typing error in the section "1.3 Satellite data and the TOEddies algorithm". The $R_{Vmax}$ is defined as follows.

Lines 221-223: "Each identified eddy was characterized by two radii associated to the two eddy boundaries defined in TOEddies: the outermost contour ($R_{out}$), and the contour with the maximum averaged geostrophic velocities ($RV_{max}$ and associated velocity $V_{max}$)."

Line 305: Please stick to "export event", export column sounds odd. Or say "the high abundance of particulate carbon in the water column…associated with the three export events"

Line 342: "export events"

Line 347: "which can be used as a proxy of water masses convergence (i.e. frontal zones)" this

should have been explained way before ! Same in Line 359.

We thank the reviewer for spotting this inconsistency. We specified it in the method in section "1.4 Lagrangian diagnostics".

Lines 242-247: "By construction, intense FTLE values (typically disposed in filament shapes) are found at the edge between two water masses that have been widely separated in the preceding days. As these two water masses come from distant locations, they are likely to share different hydrological characteristics, such as temperature, primary production or biological activity (Haller, 2015; Lehahn et al., 2018; d'Ovidio et al., 2010). For these reasons, FTLEs are useful to identify oceanic fronts and can be considered as proxies for water masses convergence and thus, possibly, associated vertical downwelling (McWilliams, 2016)"

Line 348: "consistent with the model analysis proposed by.. " you have to explain a bit better than this, or remove. What is the take-home message of this paragraph ? you suggest it is detritus from phytoplankton plus coagulation due to physical convergence ? It should be stated clearly.

We have modified the text such as:

Lines 393-396: "In other words, MaP concentrations may have been enhanced due to the combination of two important factors for coagulation according to the model of Jackson (1990), increased shear associated with high hydrodynamics at front and enhanced phytoplankton biomass"

Line 431: Statement poorly supported by results. turbulence ? It's the first time that it is used, you cannot drop that like this in the conclusion !! plus specify vertical or horizontal. I guess you want to say horizontal trapping (convergence) here ? This should be better assessed with lagrangian chl-a. Also very annoying that you are concluding about primary production although you just provide an indicator for biomass (chl-a). Also coagulation/aggregation processes are purely speculative. Or prove me wrong.

We recognise that this is also a misuse of language on our part. We modified this sentence as suggested by the reviewer. Moreover we provide more information about Lagrangian chlorophyll-a in supplementary that shows the relationship between Lagrangian chlorophyll-a and MaP in the surface.

Lines 488-489: "We propose that particle concentration at these locations is enhanced by increased primary production and/or horizontal convergence induced by frontal activity or trapping."

Line 435: abusive use of "coupling" throughout the text. Coupling is connoted. I have nothing against but would rather use "interaction" or so.

We agree with the reviewer that the word "coupling" is not suited here and we thank him/her for noting this inconsistency. We modified the sentence as follow:

Lines 493-495: "We propose that such features result from the combination of the relatively large sinking speed of such particles (likely due to their great density) and the enhanced vertical motion of the water induced by submesoscale frontal dynamics, in particular below the MLD"

Figure 8: MLD is constant throughout eddies really ? Nice but very speculative… the isopycnal slope is not very well demonstrated… I would try to provide at least more evidence of this (knowing that vertical velocities are out of reach). You can always provide some more detailed analysis in the supplement.

Having a constant depth for the MLD was for simplicity and we have now modified its depth to be qualitatively deeper in the anticyclone and shallower in the cyclone (figure 7 in the reviewed manuscript).

Line 447 : Exactly why I would have expected some quantitative numbers here.

Currently, it is not possible to achieve such a budget but in the coming decade, more results will allow it. We have modified the end of the conclusion to highlight the need for more data.

Lines 506-511: "Such studies would help to validate our proposed mechanism and to disentangle the relative contribution of the various PIPs processes (such as ESP). In particular, focus on such studies needs to be put on resolving the water movements (e.g. the vertical velocities) to enable the separation of the passive sinking and vertical translocation of particles. Since mesoscale and submesoscale structures are ubiquitous in the ocean, it would also be interesting to quantify the influence of this type of process on the overall carbon budget and to determine the extent to which it contributes to carbon sequestration in the deep ocean, information that is crucial in the context of global change."